# Motion Dynamics Learning for Few-Shot Embodied Adaptation

**Sibo He** [1]  **Weiying Xie** [1]  **Daixun Li** [1]  **Junhao Zhong** [1]  **Jiayun Tian** [1]  **Yunke Wang** [2]  **Leyuan Fang** [3]  **Gang He** [1]  **Yunsong Li** [1]

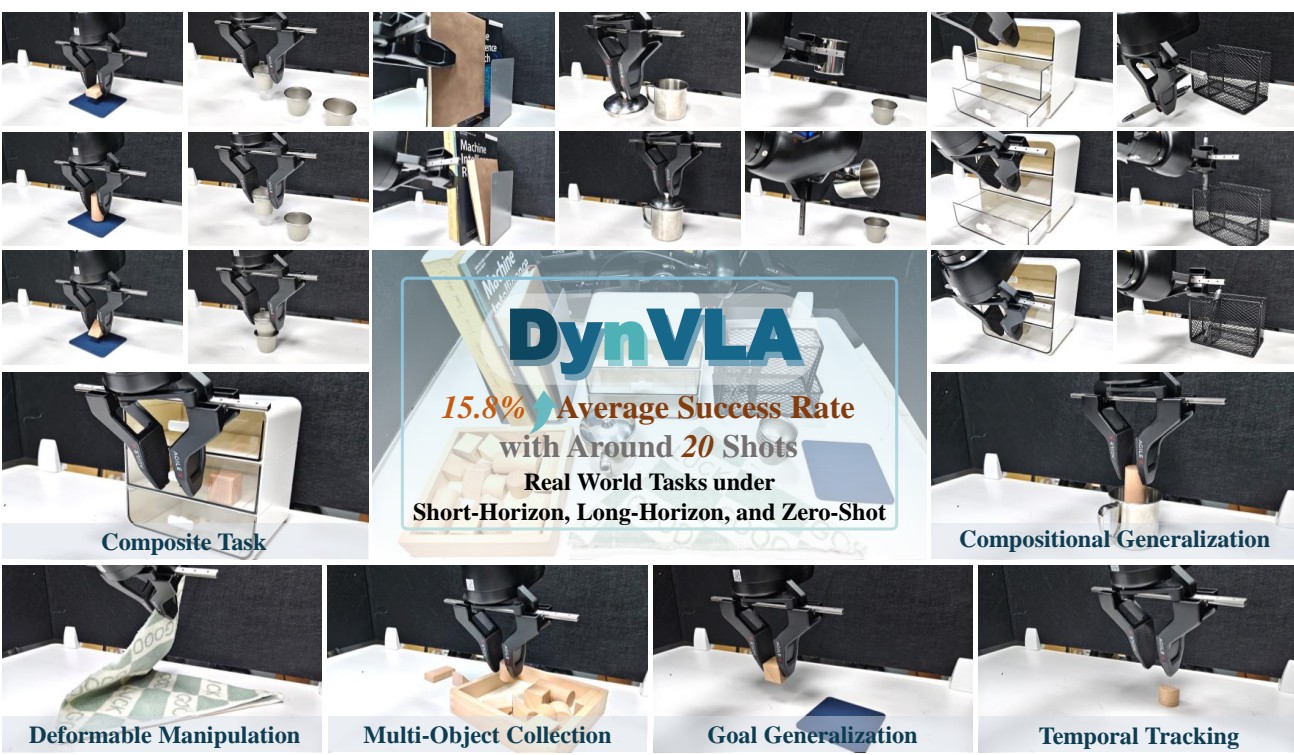

**Figure 1.** We propose DynVLA for few-shot embodied adaptation. With around 20 demonstrations for each training task, DynVLA achieves a 15.8% average success-rate improvement over the strongest baseline across 13 real-world tasks, covering short-horizon, long-horizon, and zero-shot scenarios.

## Abstract

Vision-Language-Action (VLA) models have shown strong potential for robotic manipulation, yet adapting pretrained models to novel tasks typically relies on substantial task-specific demonstrations, limiting scalability. Current VLA methods mostly focus on action imitation, which ignores the richer structure contained in trajectories. In contrast, motion dynamics governing how actions evolve over time are more informative and transferable, making them better suited for few-shot adaptation. Motivated by this idea, we propose DynVLA, a few-shot adaptation system that reformulates VLA learning from action imitation to trajectory-level motion dynamics modeling. Specifically, we propose Motion Dynamics Mechanism (MDM), which distills latent embeddings from trajectories via flow-matching inversion, yielding compact representations that capture dynamics. We further design Dynamics-Constrained Modeling (DCM). DCM projects these inferred representations onto a Dynamics Bank, which stores prior motion knowledge pretrained from diverse demonstrations. By grounding action generation in these learned priors, the system enables interpolating between existing action paradigms to represent novel dynam-

[1]Xidian University, Xi'an, China [2]The University of Sydney, Sydney, Australia [3]Hunan University, Changsha, China. Correspondence to: Weiying Xie <wyxie@xidian.edu.cn>.

*Proceedings of the 43rd International Conference on Machine Learning*, Seoul, South Korea. PMLR 306, 2026. Copyright 2026 by the author(s).

ics modes. Experiments on 13 real-world tasks demonstrate that DynVLA outperforms existing SOTA systems by 15.8% in average success rate with around 20 demonstrations, highlighting its adaptation capabilities in real-world scenes. Our code is available at https://github.com/trantor2nd/Motion-DynVLA.

# 1. Introduction

Vision-Language-Action (VLA) models (Zhou et al., 2025b; Yang et al., 2025b; Zhao et al., 2025b; Pei et al., 2026) are emerging as a new paradigm for embodied intelligence, integrating visual perception, linguistic reasoning, and action generation within a unified system (Huang et al., 2025b; Zhang et al., 2025a;b; Shi et al., 2025; Wen et al., 2025; Shukor et al., 2025). Unlike linguistic generation, robotic outputs must satisfy complex physical and geometric constraints (Zhang et al., 2025c; Liu et al., 2025a; Xu et al., 2025b). As a result, even with large-scale pretraining, real-world deployment typically still necessitates substantial task-specific data. Since such data is costly to collect, few-shot adaptation becomes a central bottleneck for practicality.

To address this problem, existing research has primarily focused on data-driven strategies such as prompt-based policy tuning (Zheng et al., 2025a; Huang et al., 2025a; Zheng et al., 2025b) or task retrieval frameworks (Yan et al., 2025; Lin et al., 2024; Xie et al., 2025a), which utilize external demonstrations to mitigate data scarcity. In these paradigms, the additional imitation objective fails to capture specific physical properties (Su et al., 2025; Li et al., 2025f;g). This often leads to the generation of trajectories that violate basic environmental constraints or geometric consistencies, especially when provided with only a handful of demonstrations (Liu et al., 2025c; Black et al., 2025b). Thus the critical bottleneck in achieving robust few-shot adaptation lies in the absence of an explicit mechanism to bridge the profound disconnect between high-level semantic reasoning and low-level motion dynamics (Xue et al., 2025; Zheng & Mei, 2025; Xie et al., 2025b; Xu et al., 2026). Rather than relying on frame-wise imitation, the path toward truly scalable embodied intelligence requires the enforcement of relevant dynamic constraints. This can be achieved by identifying and leveraging the stable motion patterns, which already reside within the high-dimensional action space of the pretrained model. By grounding the generative process in these latent dynamics, we can ensure that motion synthesis remains physically plausible and temporally consistent.

Inspired by this insight, as shown in Figure 1, we propose DynVLA, a dynamics-constrained few-shot model that formulates embodied adaptation as the inference and reuse of trajectory-level motion patterns. DynVLA intro-

duces a Motion Dynamics Mechanism (MDM) that utilizes flow-matching inversion to infer latent dynamics representations. By inverting the generative flow, MDM learns compact embeddings that parameterize the observable effects of unobserved execution factors on trajectory evolution, rather than directly measuring or disentangling explicit physical quantities. Unlike previous methods that inject task-specific identifiers or external labels, DynVLA extracts implicit embodied priors from the model itself and grounds them in a discrete Dynamics Bank consisting of stable, pretrained motion-pattern prototypes. In addition, we present Dynamics-Constrained Modeling (DCM), projecting noisy continuous representations onto these prototypes. By grounding the synthesis phase in these learned motion-pattern priors, the policy ensures that the generated actions are anchored to physically feasible regimes while allowing the model to interpolate between known paradigms to represent novel combinations of execution dynamics. Our main contributions are summarized as follows:

- We propose DynVLA, a dynamics-constrained few-shot model where action generation is conditioned on motion-pattern prototypes retrieved from a pretrained Dynamics Bank, enabling fast few-shot adaptation without extensive policy fine-tuning.

- We propose a Motion Dynamics Mechanism (MDM) driven by flow-matching inversion, which transforms frame-wise imitation into trajectory-level modeling of action sequences, yielding latent dynamics embeddings that parameterize observable execution regularities.

- We design and evaluate a suite of 13 real-world robotic tasks spanning short-horizon, long-horizon, and zero-shot settings, demonstrating that DynVLA achieves real-world adaptation with only 20 demonstrations.

# 2. Related Works

## 2.1. Vision-Language-Action Models

VLA models formulate robotic control as direct action generation conditioned on visual observations and language instructions (Liu et al., 2025b; Li et al., 2025b; Yeh et al., 2025; Zhou et al., 2025a; Li et al., 2025a;e; Zhao et al., 2025a). To achieve versatile semantic understanding and broad task-level generalization, recent VLA research commonly adopts Vision-Language Models (VLM) as the backbone to form end-to-end action generation policies trained on large-scale robot demonstrations (Yang et al., 2025a; Gao et al., 2025; Fang et al., 2025). Representative models such as OpenVLA (Kim et al., 2024), $\pi_{0.5}$ (Black et al., 2025a), and GEN-0 (Team, 2025) show that large-scale multimodal pretraining improves cross-task performance. While these models achieve strong generalization during

pretraining, adaptation to novel real-world applications typically still requires non-trivial amounts of task-specific data in fine-tuning, limiting practicality in few-shot deployment.

## 2.2. Few-Shot Policy Learning

Few-shot learning in VLA settings aims to adapt pretrained models to new real-world tasks using only a small number of demonstrations (Zheng et al.; Xu et al., 2025a; Gu et al., 2025; Hu et al., 2025; Zhang et al., 2026; 2025d; Shukor et al., 2025). Existing approaches typically achieve this by injecting additional task-specific information into large VLA policies (Driess et al., 2025; Li et al., 2025d). ControlVLA (Li et al., 2025c) introduces additional pretrained control modules and fine-tunes them with limited demonstrations to incorporate object-centric task knowledge. Retrieval-based methods (Lin et al., 2024; Xie et al., 2025a) improve few-shot imitation by constraining training data to selected prior demonstrations that are deemed relevant to the target task. Inference-time approaches (Li et al., 2025a; Kwok et al., 2025; Huang et al., 2025c) instead restrict action generation at execution time through search or evolutionary refinement without updating model parameters. While effective, these methods adapt behavior by adding task-specific knowledge or incurring increased optimization time, rather than by conditioning action generation on reusable motion dynamics, motivating our dynamics-aware approach for few-shot adaptation.

## 3. Preliminary

Advanced VLA models typically combine a VLM and a Diffusion Transformer (DiT) to process multimodal information from text, images, and states, using these as conditioning inputs for generating sequential actions.

**Perception and Reasoning.** At each execution step, the model receives a language instruction $l$, a set of multi-view RGB observations $\mathbf{I} = \{I_1, I_2, \dots\}$, and the current robot state $\mathbf{S} = \{s_1, s_2, \dots\}$, where each $s_i$ denotes an absolute joint-state value, such as the angle of an arm joint or the gripper opening state. The VLM encodes $(\mathbf{I}, l)$ into a multimodal feature representation that grounds the instruction in the current visual context, while a state encoder maps $\mathbf{S}$ into a compact proprioceptive embedding. These features are combined to form a multimodal context representation $f_c$, which is used to condition action generation.

**Action Generation.** The DiT is implemented as the policy controller, where the multimodal condition representation $f_c$ and the timestep embedding $\tau$ are integrated through standard conditioning mechanisms (Peebles & Xie, 2023).

Given a ground-truth action chunk $A_H = \{a_1, \dots, a_H\}$ with a chunk length $H$, we define the flow-matching timestep $\tau \in [0, 1]$, and the sampled Gaussian noise

$\epsilon \sim \mathcal{N}(0, I)$. The noised action chunk is computed as:

$$A_H^\tau = \tau A_H + (1 - \tau)\epsilon. \tag{1}$$

The controller prediction $\pi_\theta(f_c, A_H^\tau, \tau)$ aims to approximate the denoising vector field $(\epsilon - A_H)$ by minimizing the flow-matching objective:

$$\mathcal{L}_{\text{flow}}(\theta) = \mathbb{E}_{\tau, \epsilon} \left[ \|\pi_\theta(f_c, A_H^\tau, \tau) - (\epsilon - A_H)\|_2^2 \right]. \tag{2}$$

During inference, action chunks are generated with $K$-step denoising via forward Euler integration:

$$A_H^{\tau+1/K} = A_H^\tau + \tfrac{1}{K} \pi_\theta(f_c, A_H^\tau, \tau). \tag{3}$$

Empirically, a small number of denoising steps ($K \leq 10$) is sufficient to produce stable and coherent control trajectories.

## 4. Methodology

DynVLA is designed as a dynamics-constrained model that achieves few-shot adaptation by conditioning action generation on reusable motion-pattern prototypes. The model circumvents the need for extensive policy fine-tuning by leveraging a pretrained repository of trajectory-level execution patterns. As shown in Figure 2, our approach consists of two primary stages: (1) Motion Dynamics Mechanism (MDM), which employs flow-matching (FM) inversion and temporal contrastive learning to infer a latent dynamics representation $r_d$ from action sequences; and (2) Dynamics-Constrained Modeling (DCM), which leverages a pretrained prototype bank to retrieve and integrate these representations back into the action generation flow.

### 4.1. Motion Dynamics Mechanism

The MDM is designed to shift from frame-wise imitation towards trajectory-level dynamics modeling. The core objective is to infer a stable latent dynamics representation $r_d$ that parameterizes trajectory-level transition regularities in a motion sequence.

#### 4.1.1. DYNAMICS IDENTIFICATION VIA FM INVERSION

To bridge the gap between static multimodal perception and the dynamic requirements of action execution, we formulate the discovery of dynamics representations as a generative inverse problem. The ground-truth action sequence $A_H$ is governed by a conditional probability path $p_t(A_H | f_c, \xi)$, where $\xi$ denotes unobserved execution factors that influence trajectory evolution. These factors are not directly observed or manually disentangled; their effects are instead reflected in observable motion behaviors such as speed profiles, smoothness, contact interactions, and task-dependent transition patterns. Accordingly, MDM does not estimate $\xi$ explicitly, but infers a latent representation $r_d$ that parameterizes trajectory-level execution regularities associated with

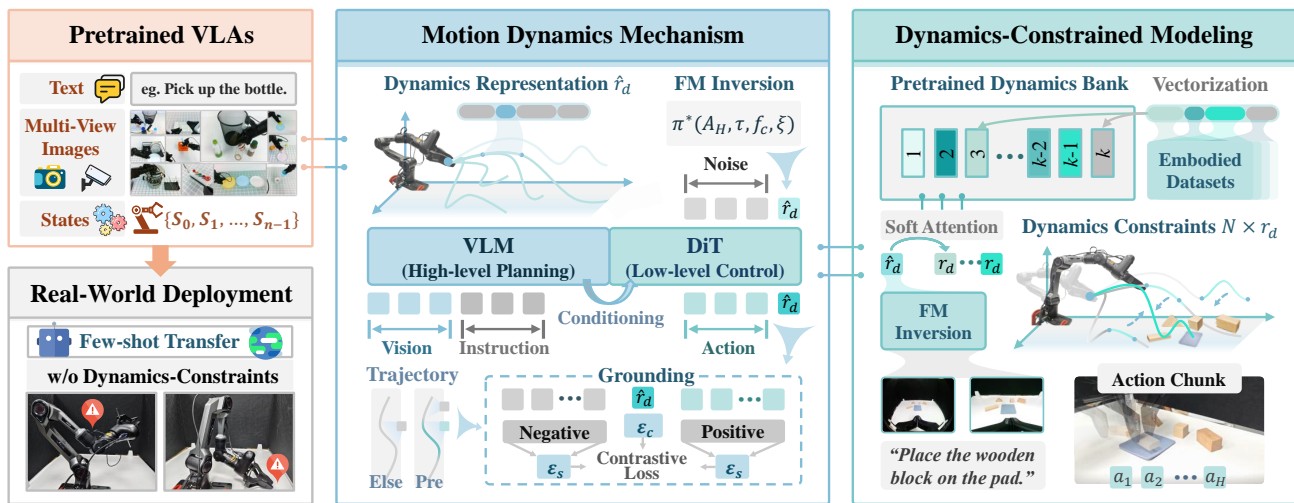

*Figure 2.* Overview of DynVLA. (Left) Pretrained VLAs struggle to bridge high-level semantics and low-level motion dynamics with limited data, leading to execution failures and lack of physical robustness in real-world deployment. (Middle) Motion Dynamics Mechanism (MDM) distills a coherent dynamics representation $\hat{r}_d$ by leveraging flow-matching inversion and temporal contrastive grounding to extract latent physical priors. (Right) Dynamics-Constrained Modeling (DCM) projects $\hat{r}_d$ onto a pretrained Dynamics Bank via soft-attention. The retrieved paradigms serve as structural anchors for the DiT to generate physically consistent action chunks.

these unobserved factors. This representation complements the context embedding $f_c$: while $f_c$ captures high-level visual-language intent, $r_d$ provides the execution structure needed to realize the action sequence $A_H$.

We introduce $r_d$ as a latent conditioning embedding within the DiT sequence. Leveraging the foundational motion priors learned in the pretrained backbone $\theta$, we define the following flow-inversion objective to optimize $r_d$:

$$\mathcal{L}_{\text{fm-inv}}(r_d) = \mathbb{E}_{\tau,\epsilon} \left[ \|\pi_\theta(f_c, A_H^\tau, \tau, r_d) - (\epsilon - A_H)\|_2^2 \right],$$
(4)

where $\pi_\theta$ denotes the pretrained conditional vector field defined along the chosen interpolation path. The inversion objective provides a training signal for $r_d$ by requiring the conditioned vector field to explain the observed action trajectory under the pretrained denoising convention. Optimizing Equation (4) identifies the conditioning embedding $r_d$ that best aligns the model-induced vector field with the observed data-noise coupling:

$$\pi_\theta(f_c, A_H^\tau, \tau, r_d) \approx v_\tau(A_H^\tau | f_c, r_d),$$
(5)

where $v_\tau(\cdot | f_c, r_d)$ denotes the conditional denoising vector field associated with the path.

Although $\xi$ is not explicitly observed, it provides a useful conceptual device for interpreting why an additional latent conditioning variable is needed. Let $x$ denote an action sample along the probability path. Applying Bayes' rule to the conditional density gives

$$p_\tau(x | f_c, \xi) = \frac{p_\tau(x | f_c) \, p_\tau(\xi | x, f_c)}{p_\tau(\xi | f_c)}.$$
(6)

Taking the gradient with respect to $x$ on both sides of the log-transformed equation yields

$$\begin{aligned} \nabla_x \log p_\tau(x | f_c, \xi) = {}& \nabla_x \log p_\tau(x | f_c) \\ & + \nabla_x \log p_\tau(\xi | x, f_c) \\ & - \nabla_x \log p_\tau(\xi | f_c). \end{aligned}$$
(7)

Since $p_\tau(\xi | f_c)$ does not depend on the specific action sample $x$, the last term vanishes. This gives the density-level decomposition

$$\nabla_x \log p_\tau(x | f_c, \xi) = \underbrace{\nabla_x \log p_\tau(x | f_c)}_{\text{Task Prior}} + \underbrace{\nabla_x \log p_\tau(\xi | x, f_c)}_{\text{Execution Correction}},$$
(8)

where the task prior term reflects the action distribution explained by the visual-language context, and the execution correction term represents how unobserved execution factors are associated with the trajectory.

In this formulation, optimizing $r_d$ in Equation (4) infers a conditioning embedding under which the pretrained vector field best explains the observed trajectory. Since the pretrained model $\theta$ captures a structured family of conditional vector fields, this inversion process identifies the embedding that makes the demonstration trajectory most consistent with the learned field family. Equivalently, it can be viewed as a likelihood-style fitting objective under the model-induced conditional path:

$$r_d^* = \arg\min_{r_d} D_{\text{KL}} \left( p_{\text{data}}(A_H | f_c) \, \| \, p_\theta(A_H | f_c, r_d) \right),$$
(9)

where the data distribution marginalizes over unobserved

execution factors:

$$p_{\text{data}}(A_H|f_c) = \int p(A_H|f_c, \xi)\, p(\xi|f_c)\, d\xi. \qquad (10)$$

The resulting embedding provides a dynamics-aware representation, which is fundamental for DCM, as any mixture of these representations remains grounded in the physical feasibility of the pretrained motion prior.

### 4.1.2. GROUNDING VIA CONTRASTIVE LEARNING

The FM inversion objective in Equation (4) provides a reconstruction-based training signal for $r_d$, but this signal alone may be underdetermined. In particular, because the inversion is optimized over a randomized data-noise coupling, multiple embeddings can produce similarly low inversion loss for the same action trajectory. This ambiguity makes the inferred representation susceptible to spurious temporal correlations and unstable local optima. To make the inversion well-conditioned, we introduce a temporal contrastive grounding objective that regularizes the latent space of $r_d$. The core idea is that embeddings inferred from temporally adjacent transitions within the same trajectory should remain consistent and predictive, while embeddings associated with mismatched context transitions should be discriminative. Thus, contrastive grounding complements FM inversion by selecting representations that not only reconstruct the current action chunk, but also remain stable across trajectory evolution.

The theoretical foundation is to maximize the mutual information $I(z_c; z_{s^+})$ between the dynamics-informed context $z_c = \psi(r_d)$ and the next state-transition embedding $z_{s^+} = \phi(f_{s,t+1})$. Here, a positive pair $(z_c, z_{s^+})$ is formed by the current latent dynamics representation and the adjacent transition within the same trajectory, while negative samples $z_{s_j^-}$ are drawn from other trajectories. We aim to maximize the dependency:

$$I(z_c; z_{s^+}) = \mathbb{E}_{p(z_c, z_{s^+})} \left[ \log \frac{p(z_{s^+}|z_c)}{p(z_{s^+})} \right]. \qquad (11)$$

Directly optimizing $I(z_c; z_{s^+})$ is intractable due to the unknown density ratio $p(z_{s^+}|z_c)/p(z_{s^+})$. However, we can construct a variational lower bound by recasting density ratio estimation as a multi-sample categorical classification task. Consider a set $X = \{z_{s^+}, z_{s_1^-}, \ldots, z_{s_{N-1}^-}\}$ containing one positive sample from the same trajectory and $N-1$ negative samples from the proposal marginal $p(z_{s^+})$. The posterior probability that the $i$-th sample is the true positive is given by:

$$P(d = i|X, z_c) = \frac{\frac{p(z_{s_i}|z_c)}{p(z_{s_i})}}{\sum_{j=1}^{N} \frac{p(z_{s_j}|z_c)}{p(z_{s_j})}}. \qquad (12)$$

By substituting the intractable density ratio with a learnable critic $h(z_c, z_s) = \exp(\langle z_c, z_s \rangle / \tau_T)$, where $\tau_T$ is a temperature scaling factor, the InfoNCE objective approximates this selection process. Following Noise-Contrastive Estimation, this log-sum-exp structure provides a variational lower bound on the mutual information:

$$I(z_c; z_{s^+}) \geq \mathbb{E} \left[ \log \frac{h(z_c, z_{s^+})}{h(z_c, z_{s^+}) + \sum_{j=1}^{N-1} h(z_c, z_{s_j^-})} \right] + \log N = -\mathcal{L}_{\text{NCE}} + \log N. \qquad (13)$$

Minimizing $\mathcal{L}_{\text{NCE}}$ therefore maximizes a lower bound of $I(z_c; z_{s^+})$, encouraging $r_d$ to preserve persistent trajectory-level execution regularities while suppressing high-frequency stochastic noise. More importantly, this objective reduces the ambiguity of FM inversion: embeddings that explain the same current action chunk but fail to predict adjacent transitions, or collapse toward embeddings from mismatched trajectories, receive a higher contrastive penalty. Thus, the contrastive objective turns the inversion from a purely reconstruction-driven fitting problem into a temporally grounded representation-learning problem.

To further stabilize the latent space for DCM, we employ a bidirectional InfoNCE objective. The forward term encourages the inferred representation to predict future transitions, while the backward term encourages observed transitions to recover a consistent latent embedding. The total temporal grounding loss is defined as:

$$\mathcal{L}_{\text{temp}} = \frac{1}{2} \left( \mathcal{L}_{\text{NCE}}^{\rightarrow} + \mathcal{L}_{\text{NCE}}^{\leftarrow} \right), \qquad (14)$$

where $\mathcal{L}_{\text{NCE}}^{\rightarrow}$ enforces predictiveness from $r_d$ to adjacent state transitions, and $\mathcal{L}_{\text{NCE}}^{\leftarrow}$ enforces consistency from observed transitions back to the latent representation.

The final MDM objective combines FM inversion and temporal grounding:

$$\mathcal{L}_{\text{MDM}} = \mathcal{L}_{\text{fm-inv}} + \lambda_{\text{temp}} \mathcal{L}_{\text{temp}}, \qquad (15)$$

where $\lambda_{\text{temp}}$ balances reconstruction fidelity and temporal grounding. This joint objective encourages $r_d$ to remain both action-consistent under Equation (4) and transition-consistent under Equation (14), thereby improving the well-posedness and stability of learned dynamics representations.

To prevent representation collapse, the state projector $\phi$ is updated via an Exponential Moving Average:

$$\hat{\theta}_{\phi}^{(t+1)} = m\, \hat{\theta}_{\phi}^{(t)} + (1 - m)\, \theta_{\phi}^{(t+1)}, \qquad (16)$$

where $m \in [0, 1)$ is the momentum coefficient. By filtering out stochastic execution noise and enforcing temporal discriminability, the learned embedding becomes a stable trajectory-level representation that can be subsequently projected onto the Dynamics Bank in DCM.

*Table 1.* Overview of the proposed VLA task suite. Tasks are grouped by category, task logic, and instructions. Shaded columns (Medium, Large, Full) represent extended data scales used to evaluate the scaling behavior and performance upper bound of policies.

| Category | Task Logic | Instruction Text | Total Demonstrations | | | | |
|---|---|---|---|---|---|---|---|
| | | | Tiny | Small | Medium | Large | Full |
| Few-shot (Short-horizon) | (T1) Pick and place | Pick up the wooden block and place it on the pad. | 18 | 36 | 54 | 72 | 90 |
| | (T2) Stacking | Stack the cups on the table. | 10 | 20 | 50 | 70 | 90 |
| | (T3) Sliding | Close the drawer of the cabinet. | 10 | 20 | 50 | 70 | 93 |
| | (T4) Orientation | Put the gel pen on the stationery organizer. | 12 | 24 | 48 | 72 | 92 |
| | (T5) Insertion | Put the book back on the shelf. | 10 | 20 | 50 | 70 | 90 |
| | (T6) Assembly | Put the lid on the teacup. | 14 | 21 | 50 | 70 | 90 |
| | (T7) Pouring | Pour the water into the cup. | 10 | 20 | 50 | 70 | 90 |
| Few-shot (Long-horizon) | (T8) Deformable manipulation | Fold the towel on the table. | 10 | 20 | 50 | 70 | 90 |
| | (T9) Composite task | Put the wooden block into a drawer and close it. | 10 | 20 | 50 | 70 | 90 |
| | (T10) Multi-object collection | Collect all the different-shaped wooden blocks. | 10 | 20 | 50 | 70 | 90 |
| Zero-shot | (T11) Goal generalization | Move the wooden block from the pad to the table. | 0 | 0 | 0 | 0 | 0 |
| | (T12) Temporal tracking | Track the movement of the wooden block in real time. | 0 | 0 | 0 | 0 | 0 |
| | (T13) Compositional generalization | Place the wooden block in the teacup. | 0 | 0 | 0 | 0 | 0 |

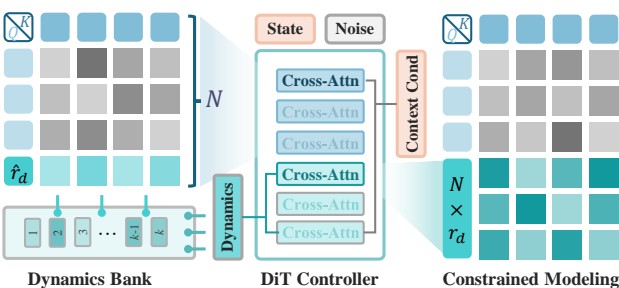

*Figure 3.* Illustration of Dynamics-Constrained Modeling. The policy extracts $\hat{r}_d$ from initial layers to retrieve motion-pattern prototypes from a discrete Dynamics Bank. These prototypes act as latent constraints for action generation in subsequent layers, grounding the trajectory within an inferred dynamics mode.

## 4.2. Dynamics-Constrained Modeling

The continuous representation $r_d$ inferred via MDM parameterizes trajectory-level execution regularities, yet its direct use in few-shot scenarios can be affected by stochasticity in real-world demonstrations. To bridge this gap, we introduce DCM, which regularizes action synthesis by leveraging a Dynamics Bank $\mathcal{P} = \{p_k\}_{k=1}^K$. Structured as a learnable prototype codebook $\mathcal{P} \in \mathbb{R}^{K \times D}$, the bank stores motion-pattern prototypes learned from diverse demonstrations. By mapping the noisy continuous estimate $\hat{r}_d$ onto these prototypes, DCM provides reusable execution-pattern anchors for action generation.

To operationalize these prototype-level constraints, we employ hierarchy-constrained conditioning within the DiT controller, executing a sequential generative pass as illustrated in Figure 3. During the initial identification phase consisting of $N$ layers, the backbone processes the current context $f_c$ and noisy action tokens $A_H^\tau$ together with a latent identification prefix. The hidden states of this prefix are projected into a set of layer-wise dynamics queries $\{r_{d,i}\}_{i=1}^N$, which summarize execution-related information at different abstraction levels. Upon reaching the architecture midpoint, each layer-

wise query $r_{d,i}$ is projected onto the Dynamics Bank to retrieve a corresponding prototype $p_i^*$ via a soft-attention mechanism:

$$p_i^* = \sum_{k=1}^K \frac{\exp(\langle r_{d,i}, p_k \rangle / \tau_B)}{\sum_{j=1}^K \exp(\langle r_{d,i}, p_j \rangle / \tau_B)} p_k, \qquad (17)$$

where $\tau_B$ is the bank-retrieval temperature. This soft retrieval produces a sequence of prototype-conditioned embeddings $p^* = \{p_i^*\}_{i=1}^N$. Since each $p_i^*$ is a convex combination of bank entries, the model can interpolate among learned motion-pattern prototypes rather than selecting a single discrete mode. This soft projection regularizes the noisy continuous estimate $\hat{r}_d$ while preserving flexibility for novel combinations of execution dynamics.

In the subsequent constrained synthesis phase, the identification prefix is replaced by the retrieved prototype sequence $p^*$, which serves as distributed conditioning tokens for the remaining DiT layers. In this formulation, $p^*$ is a prototype-space representation of the execution pattern inferred from $r_d$, rather than an explicit estimate of the latent factors $\xi$. Consequently, the conditioned vector field is written as $\pi_\theta(f_c, A_H^\tau, \tau, p^*)$. Through this prototype-level conditioning, the denoising process is biased toward reusable motion patterns learned from prior demonstrations while still allowing interpolation across prototypes.

## 5. Experiments

### 5.1. Data Acquisition

**Robot.** The experiments utilize the AgileX Piper, a 6-DOF robotic arm equipped with a Pika gripper. The vision system is configured in a dual egocentric setup: a fisheye camera captures a wide-angle scene context, while a RealSense RGB camera provides a standard perspective for precise manipulation. This combination ensures the policy perceives both global workspace layout and local object details.

**Task and Data.** The task suite is categorized into three

*Table 2.* Detailed performance on 13 tasks. Success counts ($S/20$) and categorical rates (%) are reported. **DynVLA** achieves a 73.1% Avg. at Small scale, outperforming the best baseline (GR00T-N1.5, 57.3%) by 15.8%.

| Scale | Model | Short-Horizon (T1-T7) | | | | | | | Long-Horizon (T8-T10) | | | Zero-Shot (T11-T13) | | | Success Rate (%) ↑ | | | |
|---|---|---|---|---|---|---|---|---|---|---|---|---|---|---|---|---|---|---|
| | | T1 | T2 | T3 | T4 | T5 | T6 | T7 | T8 | T9 | T10 | T11 | T12 | T13 | **Short** | **Long** | **Zero** | **Avg.** |
| **Tiny** | Diffusion Policy | 6/20 | 2/20 | 3/20 | 0/20 | 3/20 | 1/20 | 3/20 | 0/20 | 0/20 | 1/20 | 0/20 | 1/20 | 0/20 | 12.9 | 1.7 | 1.7 | 7.7 |
| | ACT | 9/20 | 4/20 | 5/20 | 1/20 | 5/20 | 2/20 | 4/20 | 1/20 | 1/20 | 2/20 | 0/20 | 2/20 | 2/20 | 21.4 | 6.7 | 6.7 | 14.6 |
| | $\pi_{0.5}$ | 11/20 | 6/20 | 8/20 | 2/20 | 8/20 | 4/20 | 8/20 | 1/20 | 3/20 | 5/20 | 2/20 | 6/20 | 5/20 | 33.6 | 15.0 | 21.7 | 26.5 |
| | GR00T-N1 | 11/20 | 6/20 | 8/20 | 2/20 | 8/20 | 4/20 | 8/20 | 1/20 | 2/20 | 5/20 | 2/20 | 5/20 | 5/20 | 33.6 | 13.3 | 20.0 | 25.8 |
| | GR00T-N1.5 | 14/20 | 8/20 | 10/20 | 3/20 | 10/20 | 5/20 | 8/20 | 3/20 | 4/20 | 7/20 | 4/20 | 7/20 | 6/20 | 41.4 | 23.3 | 28.3 | 34.2 |
| | **DynVLA (Ours)** | **16/20** | **11/20** | **12/20** | **5/20** | **12/20** | **7/20** | **13/20** | **3/20** | **6/20** | **8/20** | **2/20** | **11/20** | **10/20** | **54.3** | **28.3** | **38.3** | **44.6** |
| **Small** | Diffusion Policy | 9/20 | 5/20 | 7/20 | 2/20 | 7/20 | 4/20 | 7/20 | 1/20 | 2/20 | 4/20 | 0/20 | 2/20 | 0/20 | 29.3 | 11.7 | 3.3 | 19.2 |
| | ACT | 13/20 | 8/20 | 10/20 | 4/20 | 10/20 | 7/20 | 8/20 | 2/20 | 4/20 | 6/20 | 0/20 | 4/20 | 3/20 | 42.9 | 20.0 | 11.7 | 30.4 |
| | $\pi_{0.5}$ | 16/20 | 13/20 | 14/20 | 6/20 | 14/20 | 9/20 | 14/20 | 5/20 | 7/20 | 11/20 | 4/20 | 9/20 | 7/20 | 61.4 | 38.3 | 33.3 | 49.6 |
| | GR00T-N1 | 15/20 | 11/20 | 13/20 | 5/20 | 12/20 | 9/20 | 12/20 | 5/20 | 6/20 | 10/20 | 4/20 | 9/20 | 6/20 | 55.0 | 35.0 | 31.7 | 45.0 |
| | GR00T-N1.5 | 18/20 | 15/20 | 16/20 | 8/20 | 16/20 | 12/20 | 14/20 | 7/20 | 9/20 | 11/20 | 5/20 | 11/20 | 7/20 | 70.7 | 45.0 | 38.3 | 57.3 |
| | **DynVLA (Ours)** | **19/20** | **17/20** | **18/20** | **11/20** | **18/20** | **14/20** | **17/20** | **11/20** | **14/20** | **16/20** | **7/20** | **15/20** | **13/20** | **81.4** | **68.3** | **58.3** | **73.1** |

logic levels as shown in Table 1. Short-horizon tasks (T1-T7) cover basic primitives like picking and pouring. Long-horizon tasks (T8-T10) involve multi-stage sequences and deformable object manipulation to test temporal consistency. Zero-shot tasks (T11-T13) introduce novel instructions to evaluate the generalization of the latent action space. To analyze performance across different data regimes, we partition the teleoperated demonstration data into five hierarchical scales. The Tiny and Small scales (10-20 demos) are designed for few-shot adaptation analysis, while the Medium, Large, and Full scales (up to 90+ demos) are used to evaluate the performance upper bounds of the policies.

## 5.2. Experimental Setup

We conduct a series of experiments and compare DynVLA against several state-of-the-art models, including Diffusion Policy (Chi et al., 2025), ACT (Zhao et al., 2023), $\pi_{0.5}$ (Black et al., 2025a), GR00T-N1 (Bjorck et al., 2025), and GR00T-N1.5 (Bjorck et al., 2025). All models follow their respective pretraining strategies. DynVLA is pretrained on a curated subset of the Open X-Embodiment dataset (O'Neill et al., 2024) to acquire fundamental robotic manipulation priors. Subsequently, these models are fine-tuned on our collected task suite across the aforementioned data scales to evaluate their adaptation efficiency. The training process is conducted on 8 NVIDIA A100 (80GB) GPUs.

## 5.3. Main Result

### 5.3.1. FEW-SHOT ADAPTATION

Table 2 reports results on thirteen real-world manipulation tasks. In the Tiny and Small regimes, DynVLA achieves the highest average success, with 44.6% at Tiny and 73.1% at Small, while the strongest baseline, GR00T-N1.5, reaches 57.3% at Small. DynVLA therefore improves the Small-scale average success rate by 15.8 percentage points. On short-horizon tasks T1 to T7, $\pi_{0.5}$ and GR00T-N1.5 attain competitive success on picking, placing, and sliding, but

success decreases on tasks that require orientation control, insertion, or pouring, especially at Tiny.

The gap widens on long-horizon tasks T8 to T10, where several baselines often fail during stage transitions and show reduced continuity across multi-step execution. DynVLA improves long-horizon success by conditioning generation on retrieved dynamics constraints, which limits drift in multi-stage execution under sparse supervision. In zero-shot tasks T11 to T13, DynVLA exceeds all baselines, with 58.3% at Small versus 38.3% for GR00T-N1.5 and 33.3% for $\pi_{0.5}$, indicating better transfer to unseen instruction and goal combinations without additional training. Overall, the baseline behaviors are consistent with limited capacity to couple language with contact-rich control or to prevent error accumulation under scarce demonstrations, while DynVLA explicitly infers a dynamics representation and enforces prototype-based constraints during generation, which supports stable multi-stage execution and improves transfer.

### 5.3.2. DATA SCALING

Table 3 analyzes performance as the data scale increases from Small to Full. DynVLA already exhibits competitive performance at Small and then improves as data grows, with the overall average increasing from 73.1% at Small to 82.7% at Full. This indicates that DynVLA reaches a high capability regime earlier than other models, establishing a higher lower bound under limited demonstrations. After approaching this regime, additional data still yields measurable gains, but with diminishing increments, which is consistent with the method saturating toward a task-dependent performance boundary while continuing to refine execution quality.

In contrast, GR00T-N1.5 increases from 57.3% at Small to 72.3% at Medium and 77.3% at Full, and $\pi_{0.5}$ increases from 49.6% at Small to 66.2% at Medium and 71.5% at Full. The larger gains for these baselines suggest that they rely more directly on accumulating demonstrations to internalize task dynamics and multi-step contingencies, so per-

*Table 3.* Data scaling analysis. Improvements (∆) are relative to the **Small** scale.

| Model | Short-Horizon | | | Long-Horizon | | | Zero-Shot | | | Overall Average | | |
|---|---|---|---|---|---|---|---|---|---|---|---|---|
| | Med. | Large | Full | Med. | Large | Full | Med. | Large | Full | Med. | Large | Full |
| Diffusion Policy | 31.4 (+2.1) | 33.6 (+4.3) | 34.3 (+5.0) | 13.3 (+1.6) | 15.0 (+3.3) | 15.0 (+3.3) | 5.0 (+1.7) | 5.0 (+1.7) | 6.7 (+3.4) | 21.2 (+2.0) | 22.7 (+3.5) | 23.5 (+4.3) |
| ACT | 45.7 (+2.8) | 48.6 (+5.7) | 49.3 (+6.4) | 23.3 (+3.3) | 26.7 (+6.7) | 28.3 (+8.3) | 13.3 (+1.6) | 15.0 (+3.3) | 16.7 (+5.0) | 33.1 (+2.7) | 35.8 (+5.4) | 36.9 (+6.5) |
| $\pi_{0.5}$ | 75.7 (+14.3) | 78.6 (+17.2) | 80.0 (+18.6) | 58.3 (+20.0) | 63.3 (+25.0) | 65.0 (+26.7) | 51.7 (+18.4) | 55.0 (+21.7) | 58.3 (+25.0) | 66.2 (+16.6) | 69.6 (+20.0) | 71.5 (+21.9) |
| GR00T-N1 | 71.4 (+16.4) | 75.0 (+20.0) | 76.4 (+21.4) | 55.0 (+20.0) | 60.0 (+25.0) | 61.7 (+26.7) | 48.3 (+16.6) | 53.3 (+21.6) | 56.7 (+25.0) | 62.3 (+17.3) | 66.5 (+21.5) | 68.5 (+23.5) |
| GR00T-N1.5 | 81.4 (+10.7) | 85.0 (+14.3) | 85.7 (+15.0) | 70.0 (+25.0) | 73.3 (+28.3) | 75.0 (+30.0) | 53.3 (+15.0) | 60.0 (+21.7) | 60.0 (+21.7) | **72.3** (+15.0) | **76.5** (+19.2) | **77.3** (+20.0) |
| **DynVLA (Ours)** | **85.7** (+4.3) | **89.3** (+7.9) | **90.0** (+8.6) | **73.3** (+5.0) | **76.7** (+8.4) | **78.3** (+10.0) | **65.0** (+6.7) | **68.3** (+10.0) | **70.0** (+11.7) | **78.1** (+5.0) | **81.5** (+8.4) | **82.7** (+9.6) |

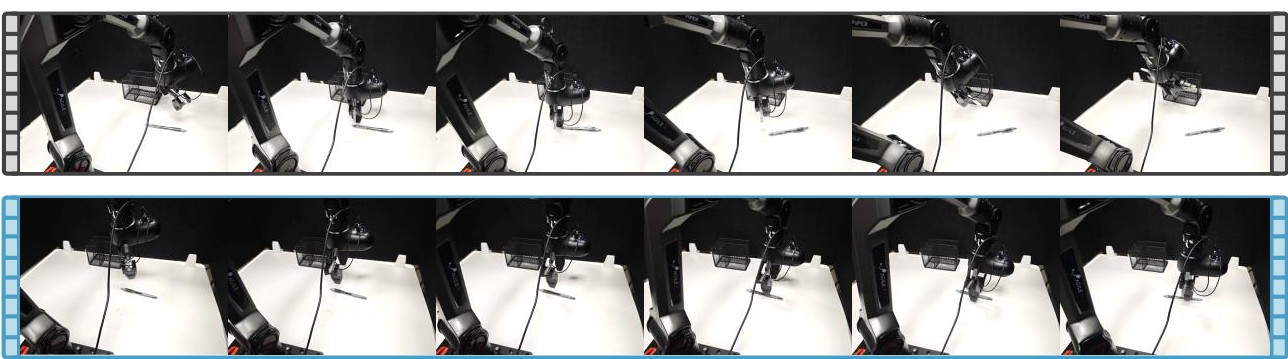

*Figure 4.* Comparative analysis of execution trajectories for the Orientation task (T4). The top sequence shows the failure mode of the GR00T-N1 baseline, and the bottom sequence shows the successful performance of DynVLA.

formance at Small reflects a weaker few-shot lower bound rather than early saturation. The category results follow the same pattern. For DynVLA, short-horizon success increases from 81.4% to 90.0%, long-horizon increases from 68.3% to 78.3%, and zero-shot increases from 58.3% to 70.0% at Full, while maintaining the lead in long-horizon and zero-shot across all scales. ACT and Diffusion Policy improve with more data but remain substantially lower on long-horizon and zero-shot, consistent with limited robustness to error accumulation and weaker coupling between instruction and contact-rich control. GR00T-N1.5 scales with data and reaches 77.3% at Full, but it still trails DynVLA by 5.4 percentage points at Full and by 15.8 percentage points at Small, suggesting that stronger pretraining alone does not establish the same few-shot lower bound. Overall, the scaling behavior supports that DynVLA reduces sensitivity to demonstration count by explicitly inferring a dynamics representation and constraining generation with retrieved prototypes, which preserves behavior at Small, and then leverages additional data to further improve performance.

### 5.3.3. QUALITATIVE FAILURE ANALYSIS

Figure 4 further illustrates the difference on the orientation-sensitive task T4. This task is challenging because the pen is initialized with a randomized yaw angle, so successful execution requires not only reaching the object but also coordinating the wrist orientation with the object's principal axis before contact. In the GR00T-N1 rollout, the end-effector approaches the pen with limited yaw adjustment, and the

gripper contacts the object from the side before establishing a stable grasp. This contact pushes the pen away from its initial position and causes the trial to fail. In the DynVLA rollout, the end-effector first rotates during the descent phase, aligns the gripper opening with the pen axis, and then closes the fingers after alignment. This comparison shows that the two policies differ most clearly in the approach-and-alignment stage, which is critical for orientation-sensitive manipulation under limited demonstrations.

### 5.4. Ablation Study

In this section, we conduct extensive ablation experiments on the **Small** scale to isolate the contributions of our core modules and analyze the sensitivity of DynVLA to key architectural hyperparameters.

#### 5.4.1. ABLATION OF MAIN DESIGN

We conduct a comparative analysis using three configurations as shown in Table 4. We compare the full DynVLA against: (1) Baseline, which removes components entirely, (2) w/o DCM, which excludes the dynamic trajectory refinement, and (3) w/o MDM, which specifically ablates the grounding via contrastive learning. In this case, we retain the few-shot inversion process to maintain the latent-based structure, as entirely removing the latent interface would cause the model to collapse back to the baseline.

The empirical results reveal that both modules are essential

*Table 4.* Ablation of main design on the **Small** scale.

| Configuration | Short (%) | Long (%) | Zero (%) | Avg. (%) |
|---|---|---|---|---|
| (1) Baseline | 55.0 | 35.0 | 31.7 | 45.0 |
| (2) w/o DCM | 68.6 | 48.3 | 45.0 | 58.5 |
| (3) w/o MDM | 61.4 | 41.7 | 38.3 | 51.5 |
| **DynVLA (Full)** | **81.4** | **68.3** | **58.3** | **73.1** |

*Table 5.* Ablation of codebook size $K$ on **Small** scale.

| Codebook Size $K$ | Short (%) | Long (%) | Zero (%) | Avg. (%) |
|---|---|---|---|---|
| 512 | 75.7 | 60.0 | 51.7 | 66.5 |
| **1024 (Default)** | **81.4** | **68.3** | **58.3** | **73.1** |
| 2048 | 80.7 | 68.3 | 56.7 | 72.3 |
| 4096 | 79.3 | 63.3 | 51.7 | 69.2 |

*Table 6.* Ablation of constraint depth $N$ on **Small** scale.

| Depth $N$ | Short (%) | Long (%) | Zero (%) | Avg. (%) |
|---|---|---|---|---|
| 2 | 57.1 | 35.0 | 33.3 | 46.5 |
| 4 | 72.1 | 51.7 | 48.3 | 61.9 |
| **6 (Default)** | **81.4** | **68.3** | **58.3** | **73.1** |
| 8 | 65.0 | 46.7 | 43.3 | 55.8 |
| 10 | 53.6 | 31.7 | 31.7 | 43.5 |

for achieving high-fidelity manipulation, but they serve distinct functional roles. Removing the grounding (w/o MDM) leads to a significant performance drop across all categories, with the largest decrease appearing in long-horizon tasks, indicating that temporal grounding is critical for producing a dynamics representation that remains stable for interpretation under limited demonstrations. On the other hand, the exclusion of the DCM (w/o DCM) causes a pronounced decline in long-horizon success rates. This indicates that injecting constraints is required to preserve temporal coherence over multi-stage execution, limiting drift that otherwise compounds into failures at stage transitions. In this interpretation, MDM yields grounded latent dynamics primitives, and DCM structures their sequential composition during generation through prototype-based conditioning.

### 5.4.2. ABLATION OF CONSTRAINT CAPACITY

This experiment investigates the sensitivity of DynVLA to the codebook size $K$, which defines the resolution of the discrete latent action space. We evaluate the performance across a range of capacities from $K = 512$ to $K = 4096$, as shown in Table 5. The results indicate that $K = 1024$ serves as the optimal bottleneck for capturing the complexity of robotic manipulation. At the lower capacity of $K = 512$, we observe a performance degradation across all metrics, with a larger drop on long-horizon tasks. This suggests that a smaller codebook lacks the expressivity required to represent fine-grained action primitives. Conversely, expanding the capacity to $K = 4096$ results in a notable decline in zero-shot generalization and long-horizon stability. This drop suggests that an excessively large codebook tends to overfit to the specific trajectory noise present in the few-shot demonstration set, thereby weakening the ability to extract generalizable dynamics representations. The stability observed at $K = 2048$ reinforces that once the latent space achieves sufficient representational richness, further increasing capacity yields diminishing returns and eventually introduces optimization challenges.

### 5.4.3. ABLATION OF CONSTRAINT DEPTH

The constraint depth $N$ specifies the layer within the DiT-based controller where the dynamic representation $r_d$ is injected to regularize trajectory generation. As shown in Table 6, performance reaches the best average success rate at $N = 6$ under the Small-scale setting across short-horizon, long-horizon, and zero-shot evaluations. Shallower settings, such as $N = 2$ and $N = 4$, yield lower averages of 46.5% and 61.9%, suggesting that injecting the conditioning signal too early is less effective in this architecture. Deeper settings also reduce performance, with 55.8% at $N = 8$ and 43.5% at $N = 10$, indicating that very late injection leaves fewer DiT layers to propagate the retrieved prototype information. Overall, these results show that a middle-layer injection point provides the best empirical trade-off.

## 6. Limitation

Current evaluation focuses on tabletop manipulation with a fixed robot platform and daily objects. We observe higher failure rates near the edge of the reachable workspace.

## 7. Conclusion

In this paper, we propose DynVLA, a dynamics-constrained vision-language-action model for few-shot embodied adaptation. Unlike existing methods, DynVLA transforms action generation into prior dynamics governed trajectory modeling. By integrating MDM to infer latent dynamics embeddings via FM inversion, and a DCM stage to anchor action generation within a pretrained Dynamics Bank, DynVLA effectively bridges the gap between high-level semantic reasoning and low-level physical consistency. Extensive evaluations across 13 real-world tasks with comprehensive settings demonstrate that DynVLA achieves superior performance, ensuring high reliability in few-shot adaptation.

## Acknowledgments

This work was supported in part by the National Natural Science Foundation of China under Grants 62531018, 62322117, 62425109, 62371365, and U24B20136, and in part by the Fundamental and Interdisciplinary Disciplines Breakthrough Plan of the Ministry of Education of China under Grant JYB2025XDXM105.

## Impact Statement

While DynVLA demonstrates strong adaptation in real-world settings, we emphasize the importance of careful deployment and continuous monitoring to ensure safe and reliable behavior in actual industrial production activities.

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

## A. Problem Analysis for Few-Shot Embodied Adaptation

Few-shot embodied adaptation poses a fundamental challenge for Vision-Language-Action (VLA) systems: while high-level semantic understanding can often be transferred from large-scale pretraining, low-level motion execution remains highly sensitive to task-specific physical dynamics. Under limited demonstrations, existing methods typically rely on frame-wise imitation or prompt-based conditioning, which implicitly assume that actions can be directly composed from sparse observations. However, such approaches fail to account for the fact that embodied tasks are governed by continuous and structured motion dynamics, where temporal consistency and physical feasibility play a central role.

This mismatch becomes particularly pronounced in real-world settings, where small deviations in contact, timing, or object properties can accumulate into execution failures. As a result, few-shot adaptation is not merely a problem of insufficient data coverage, but rather a problem of how to effectively extract and reuse latent motion patterns that are already embedded in pretrained models. These observations motivate a shift from action-level imitation toward a dynamics-centric perspective, where adaptation is achieved by identifying, grounding, and reusing motion dynamics that generalize across tasks.

## B. Platform and Real-World Setup

### B.1. Embodied Entity

As shown in Figure 5, the physical execution of the 13 real-world tasks is conducted using the AgileX Piper, a 6-DoF robotic arm. The hardware architecture features an integrated control system with a lightweight aluminum alloy skeleton and a durable plastic shell, resulting in a total mass of 4.2 kg. This lightweight design allows for the high-speed execution of trajectories while maintaining a payload capacity of 1.5 kg, which is sufficient for handling the diverse objects used in our task suite, such as wooden blocks, gel pens, and teacups.

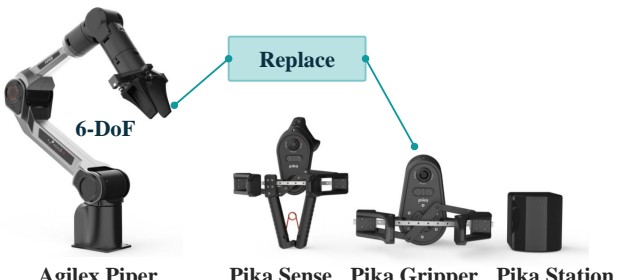

*Figure 5.* AgileX Piper and Pika.

The end-effector is equipped with a Pika parallel gripper, which is optimized for both rigid and deformable objects. The sensor suite integrated into the entity follows a dual-egocentric configuration to provide a comprehensive observation. Specifically, a wide-angle fisheye camera is utilized to capture the global workspace layout and task-level scene context, while a RealSense RGB camera provides a standard perspective focused on the gripper's local interactions.

Unlike traditional robotic setups that rely on external third-person cameras fixed to the environment, our egocentric combination entirely discards the need for complex third-person camera calibration and specialized lighting synchronization. This self-contained sensing strategy ensures that the entire system remains plug-and-play, allowing researchers to replicate our experimental results by simply placing the robot on any standard desktop without the overhead of re-aligning external sensors. The joint specifications and mechanical limits of the platform are summarized in Table 7 and Table 8 to provide a comprehensive technical reference.

### B.2. Data Collection

#### B.2.1. OPERATION MODE

The acquisition of high-quality demonstration data for the 13 real-world tasks is facilitated by two complementary operation modes, ensuring both intuitive movement and precise control. The primary mode utilizes Pika Sense teleoperation, where a human operator employs a handheld device to map natural arm and wrist motions directly to the robot's 6-DOF workspace in real-time. This mode is particularly essential for capturing the complex, contact-rich motion dynamics required for

*Table 7.* Mechanical and Technical Specifications of the AgileX Piper (PiPER) Platform.

| Mechanical Parameter | Specification Value |
| --- | --- |
| Degrees of Freedom (DOF) | 6 |
| Effective Payload | 1.5 kg |
| Structural Weight | 4.2 kg |
| Repeatability Precision | $\pm 0.1$ mm |
| Working Radius | 626.75 mm |
| Base Mounting Size | 70 mm $\times$ 70 mm $\times$ M5*4 |
| Material Composition | Aluminum Alloy and Plastic Shell |
| Communication Interface | CAN |
| Joint Range J1 (Base) | $\pm 154°$ |
| Joint Range J2 (Shoulder) | $0° \sim 195°$ |
| Joint Range J3 (Elbow) | $-175° \sim 0°$ |
| Joint Range J4 (Wrist Yaw) | $-106° \sim 106°$ |
| Joint Range J5 (Wrist Pitch) | $-75° \sim 75°$ |
| Joint Range J6 (Wrist Roll) | $\pm 100°$ |
| Maximum Joint Speed (J1-J3) | $180°/s \sim 195°/s$ |
| Maximum Joint Speed (J4-J6) | $225°/s$ |

*Table 8.* Technical Specifications of the Pika Gripper and Sensing Suite.

| Category | Parameter Project | Technical Specification |
| --- | --- | --- |
| **Pika Gripper** | Clamping Type | Two-finger Parallel Gripper |
| | Maximum Clamping Force | 2 kg |
| | Opening Range | 0 mm $\sim$ 95 mm |
| | Measurement Precision | $\pm 0.1°$ |
| | Total Weight | 690 g |
| **Depth Camera (unused)** | - | - |
| **Fisheye Camera** | Diagonal View Angle | 200° |
| | Standard Output | 1280 $\times$ 720 @ 30 FPS |
| | High-speed Output | 640 $\times$ 480 @ 90 FPS |
| **Inertial Sensor** | IMU Type | 9-axis Gyroscope |
| | Feedback Frequency | 100 Hz |

tasks such as deformable manipulation and precise assembly. For simpler task primitives or scenarios requiring discrete, structured movements, we also provide a keyboard-based control interface that allows for the execution of reproducible action sequences. These dual modes enable the rapid collection of the 10-20 high-fidelity demonstrations per task, providing a robust behavioral dataset that serves as the foundation for the system's few-shot adaptation capabilities.

### B.2.2. RANDOMIZATION SETUP

To ensure experimental fairness and spatial diversity in the collected demonstrations, we implement a systematic grid-based randomization strategy. The workspace is partitioned into a structured grid consisting of 4 rows and 9 columns, as illustrated by the crosshair markers in Figure 6. During the data collection phase, the target coordinates for object placement and the initial orientations of the items are generated stochastically. To reflect the density of typical manipulation tasks, we employ a weighted sampling distribution where coordinates closer to the center of the grid have a higher probability of occurrence compared to those at the periphery. This controlled randomization ensures that the demonstration dataset covers a comprehensive range of initial states while prioritizing the most common interaction zones, thereby establishing a rigorous and unbiased foundation for evaluating the adaptation capabilities.

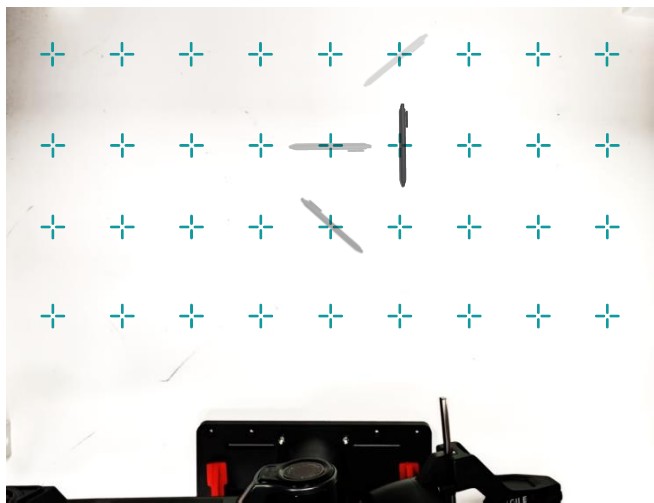

*Figure 6.* Visualization of Randomization Setup.

## B.3. Task Suite Demonstrations

### B.3.1. (T1) PICK AND PLACE.

Instruction: Pick up the wooden block and place it on the pad.

Description: The Pick and Place task serves as the foundational primitive within our suite, designed to evaluate the system's ability to coordinate basic reaching, grasping, and target-oriented transport. The workflow requires the robotic arm to identify a wooden block located at a randomized position, execute a stable top-down grasp, and subsequently move the object to be placed precisely onto a pad. This task is essential for establishing a baseline for spatial accuracy and the integration of the dual-egocentric vision system with 6-DOF action generation. The primary challenges lie in the high variance of the block's initial pose and the requirement for precise depth estimation to ensure a secure grasp and accurate release, especially given the limited training samples which make the model susceptible to compounding spatial errors during the transport phase. (Figure 7 and Figure 8)

### B.3.2. (T2) STACKING

Instruction: Stack the cups on the table.

Description: The Stacking task is designed to evaluate the system's precision in vertical alignment and its capacity for multi-object spatial reasoning. The procedure involves the robot identifying a cup, executing a secure grasp, and then precisely aligning its center with the mouth of a target cup to form a stable stack. A minor horizontal deviation during the release phase can cause the stack to topple, requiring the model to maintain strict axial consistency. (Figure 9 and Figure 10)

### B.3.3. (T3) SLIDING

Instruction: Close the drawer of the cabinet.

Description: The Sliding task is incorporated into the suite to evaluate the system's ability to perform movements under external mechanical constraints, where the object's motion is restricted to a single translational axis. The experimental setup consists of a desktop storage cabinet with a drawer initially pulled out to a random distance. The procedure requires the robotic arm to approach the drawer handle or front face and apply a consistent horizontal force to slide the component back into its housing until it is completely flush with the cabinet frame. This task differs from free-space manipulation as it necessitates maintaining a precise force vector aligned with the drawer's physical slide rails. The primary difficulties involve the risk of mechanical jamming if the applied force deviates from the sliding axis. (Figure 11 and Figure 12)

### B.3.4. (T4) ORIENTATION

Instruction: Put the gel pen on the stationery organizer.

Description: This task is designed to evaluate the system's dexterity for thin, elongated objects under high spatial uncertainty. The experimental setup involves a gel pen placed on the workspace with a completely randomized initial yaw angle and a wire mesh stationery organizer. The procedure requires the robotic arm to approach the pen from an arbitrary direction, execute a stable grasp on the cylindrical body, and subsequently rotate the wrist to restore a horizontal visual perspective before proceeding to the placement phase. A major difficulty of this task resides in the initial grasping stage, where the narrow diameter of the pen and the limited contact surface of the parallel gripper allow for almost no tolerance in angular alignment. Furthermore, because the vision system is mounted on the wrist of the robot, the act of aligning the gripper with the orientation of the pen results in a significantly tilted and non-standard camera view. The model must successfully process these distorted visual perspectives and learn the necessary dynamics to rotate the coordinate frame back to a level horizon, which is essential for the stable movement and accurate placement of the object into the organizer. (Figure 13 and Figure 14)

### B.3.5. (T5) INSERTION

Instruction: Put the book back on the shelf.

Description: The Insertion task is designed to evaluate the capability of the system to manage narrow-space alignment under severe visual constraints. The setup consists of a book on the table and a shelf with a vertical gap formed by other books. The procedure requires the robot to grasp the object and maneuver it into the slot, perfectly parallel to the shelf plane before entry. Furthermore, as the robot approaches the shelf, the volume of the held book significantly obstructs the field of view of the egocentric cameras. This creates a blind spot during the most critical phase of the maneuver, forcing the model to rely on the motion patterns established during the initial approach to maintain a stable trajectory when the target insertion point is no longer visible. (Figure 15 and Figure 16)

### B.3.6. (T6) ASSEMBLY

Instruction: Put the lid on the teacup.

Description: This involves the coordination required to join two separate components with fitting geometries. The experimental setup consists of a metallic teacup and a corresponding rounded lid placed on the workspace. The procedure requires the robotic arm to grasp the lid and place it onto the teacup to complete the assembly. The lid is a small object with a limited and curved grasping surface, requiring accurate finger placement to prevent mechanical slippage. During the final assembly phase, the egocentric cameras experience significant occlusion because the gripper and the lid block the line of sight to the rim of the teacup. Successful execution relies on the ability to maintain the centered trajectory of the lid based on the spatial relationship established before the target becomes obscured. (Figure 17 and Figure 18)

### B.3.7. (T7) POURING

Instruction: Pour the water into the cup.

Description: This task involves the transfer of contents between two containers, requiring complex rotational coordination. The experimental setup consists of a target metallic cup and a smaller source container placed on the workspace. The procedure requires the robot to grasp the source container, move it above the target, and tilt the wrist to execute the pour. The task requires maintaining the lip directly over the target opening during the tilting phase. Additionally, the tilting motion results in a significant rotation of the egocentric camera view, forcing the model to process a rapidly changing visual coordinate frame while maintaining the spatial alignment necessary for a successful pour. (Figure 19 and Figure 20)

### B.3.8. (T8) DEFORMABLE MANIPULATION

Instruction: Fold the towel on the table.

Description: The Deformable Manipulation task serves as a representative long-horizon challenge. The experimental setup involves a towel placed on a flat surface. The procedure requires the robotic arm to identify a specific corner, execute a pinch grasp, and perform a multi-stage trajectory to fold one section of the fabric over another. A fundamental difficulty of this task resides in the stochastic nature of deformable materials, which can crumple or shift in unpredictable ways, necessitating high temporal consistency to maintain a stable folding motion. Furthermore, a critical challenge involves the ability of the system to relocate key grasping points after the initial deformation. Because the visual features and geometric configuration of the fabric change significantly during the folding process, the model must successfully track and identify valid interaction points on a non-rigid surface that no longer matches the initial state. (Figure 21 and Figure 22)

### B.3.9. (T9) COMPOSITE TASK

Instruction: Put the wooden block into a drawer and close it.

Description: The Composite Task is a complex long-horizon challenge designed to test the multi-stage reasoning and execution capabilities of the system. The environment consists of a wooden block and a storage cabinet with an open drawer. The procedure involves a sequential workflow where the robot must first locate and grasp the wooden block, transport it to the cabinet, place it inside the open drawer, and finally perform a sliding maneuver to close the drawer completely. This task requires the seamless integration of different manipulation primitives, transitioning from free-space transport to constrained-axis sliding. (Figure 23 and Figure 24)

### B.3.10. (T10) MULTI-OBJECT COLLECTION

Instruction: Collect all the different-shaped wooden blocks.

Description: The Multi-object Collection task is a long-horizon challenge that focuses on the management of multiple items within a cluttered workspace. The experimental setup consists of various wooden blocks of different geometric shapes, including cylinders and prisms, distributed across the table near a collection tray. The procedure requires the robotic arm to sequentially identify, grasp, and transport every individual block into the tray until the table surface is clear. One significant difficulty of this task resides in the geometric diversity of the objects, which requires the adjustment of approach angles and grasping widths for each specific shape. Furthermore, the presence of multiple objects in close proximity increases the likelihood of accidental collisions during the movement of the gripper. (Figure 25 and Figure 26)

### B.4. Evaluation Protocol

The evaluation is conducted using a rigorous randomized testing protocol designed to assess the boundary of the generalization of the model. To ensure that the performance metrics reflect true adaptation rather than trajectory memorization, all test trials are executed at locations within the interstitial areas between the specific coordinates used during the demonstration collection phase (Figure 6). In the context of real-world robotics, twenty trials per task constitute a standard benchmark that allows for the calculation of performance gaps with sufficient granularity to distinguish between state-of-the-art baselines without incurring excessive hardware fatigue.

The evaluation suite further incorporates three distinct zero-shot tasks:

(T11) Goal Generalization, evaluates the ability of the system to maneuver objects to target locations that reside entirely outside the spatial grid defined during training.

(T12) Temporal Tracking, assesses the robustness of the action generation when the target object is dynamically relocated by an external agent during the execution of the task, requiring the model to update its trajectory in real-time.

(T13) Compositional Generalization, requires the model to synthesize a known primitive with a novel object-container combination that was not part of the training data for that specific logic.

These zero-shot benchmarks serve as a stringent verification of the capacity of the system to utilize its pretrained priors to follow environmental changes and novel instructions without the requirement for additional data.

## C. Implementation Details

The fine-tuning process is conducted using a standardized set of hyperparameters listed in Table 9.

## D. Detailed Experimental Results

### D.1. Explanation of Baseline Selection

The evaluation of DynVLA is conducted against five state-of-the-art baselines that represent the primary paradigms in modern robotic manipulation learning. Diffusion Policy (Chi et al., 2025) is selected to represent the standard for modeling multimodal action distributions through diffusion processes, while Action Chunking Transformer (ACT) (Zhao et al., 2023) provides a benchmark for high-precision control and temporal coherence via transformer-based action chunking. These two models serve as representatives of specialized visuomotor policies that rely on task-specific data.

*Table 9.* Key Hyperparameters for the Fine-Tuning of DynVLA.

| Parameter Project | Setting / Value |
|---|---|
| Batch Size | 32 |
| Action Chunk Size ($H$) | 50 |
| Observation Steps | 1 |
| Numerical Precision | bfloat16 (BF16) |
| Optimizer Type | AdamW |
| Peak Learning Rate | $1 \times 10^{-4}$ |
| Weight Decay | $1 \times 10^{-5}$ |
| Adam Betas ($\beta_1, \beta_2$) | (0.95, 0.999) |
| Gradient Clipping Norm | 10.0 |
| LR Scheduler | Cosine Decay with Warmup |
| Warmup Steps | 500 |
| Maximum Training Steps | 30,000 |

In addition, we evaluate DynVLA against a suite of generalist VLA models, namely $\pi_{0.5}$ (Black et al., 2025a), GR00T-N1 (Bjorck et al., 2025), and GR00T-N1.5 (Bjorck et al., 2025). Specifically, $\pi_{0.5}$ represents large-scale policies trained on diverse open-world datasets, GR00T-N1 embodies recent advancements in generalist policies for multi-embodiment transfer, and GR00T-N1.5 serves as a stronger upgraded generalist baseline. Collectively, these systems allow for a comprehensive assessment of how the dynamics-constrained mechanism in DynVLA performs relative to models utilizing pure action imitation, specialized visuomotor architectures, or large-scale data pretraining.

## D.2. Results of Data Expansion

*Table 10.* Expanded scaling results on 13 tasks for Med., Large, and Full scales. Success counts ($S/20$) and categorical rates (%) are reported. All averages are task-weighted to ensure strict consistency with the scaling analysis in the main text.

| Scale | Model | Short-Horizon (T1-T7) | | | | | | | Long-Horizon (T8-T10) | | | Zero-Shot (T11-T13) | | | Success Rate (%) ↑ | | | |
|---|---|---|---|---|---|---|---|---|---|---|---|---|---|---|---|---|---|---|
| | | T1 | T2 | T3 | T4 | T5 | T6 | T7 | T8 | T9 | T10 | T11 | T12 | T13 | Short | Long | Zero | Avg. |
| **Med.** | Diffusion Policy | 10/20 | 6/20 | 8/20 | 2/20 | 7/20 | 4/20 | 7/20 | 2/20 | 2/20 | 4/20 | 0/20 | 2/20 | 1/20 | 31.4 | 13.3 | 5.0 | 21.2 |
| | ACT | 14/20 | 9/20 | 11/20 | 5/20 | 10/20 | 7/20 | 8/20 | 3/20 | 5/20 | 6/20 | 1/20 | 4/20 | 3/20 | 45.7 | 23.3 | 13.3 | 33.1 |
| | $\pi_{0.5}$ | 18/20 | 16/20 | 17/20 | 10/20 | 16/20 | 13/20 | 16/20 | 9/20 | 11/20 | 15/20 | 8/20 | 12/20 | 11/20 | 75.7 | 58.3 | 51.7 | 66.2 |
| | GR00T-N1 | 17/20 | 15/20 | 16/20 | 10/20 | 15/20 | 12/20 | 15/20 | 9/20 | 10/20 | 14/20 | 7/20 | 12/20 | 10/20 | 71.4 | 55.0 | 48.3 | 62.3 |
| | GR00T-N1.5 | 19/20 | 17/20 | 18/20 | 11/20 | 17/20 | 14/20 | 18/20 | 12/20 | 14/20 | 16/20 | 9/20 | 12/20 | 11/20 | 81.4 | 70.0 | 53.3 | 72.3 |
| | **DynVLA (Ours)** | **20/20** | **18/20** | **19/20** | **13/20** | **19/20** | **15/20** | **16/20** | **13/20** | **15/20** | **16/20** | **11/20** | **14/20** | **14/20** | **85.7** | **73.3** | **65.0** | **78.1** |
| **Large** | Diffusion Policy | 11/20 | 7/20 | 8/20 | 2/20 | 8/20 | 4/20 | 7/20 | 2/20 | 3/20 | 4/20 | 0/20 | 2/20 | 1/20 | 33.6 | 15.0 | 5.0 | 22.7 |
| | ACT | 15/20 | 10/20 | 12/20 | 5/20 | 11/20 | 7/20 | 8/20 | 3/20 | 6/20 | 7/20 | 1/20 | 5/20 | 3/20 | 48.6 | 26.7 | 15.0 | 35.8 |
| | $\pi_{0.5}$ | 19/20 | 16/20 | 18/20 | 11/20 | 17/20 | 14/20 | 15/20 | 10/20 | 12/20 | 16/20 | 9/20 | 13/20 | 11/20 | 78.6 | 63.3 | 55.0 | 69.6 |
| | GR00T-N1 | 18/20 | 16/20 | 17/20 | 10/20 | 16/20 | 13/20 | 15/20 | 10/20 | 11/20 | 15/20 | 8/20 | 13/20 | 11/20 | 75.0 | 60.0 | 53.3 | 66.5 |
| | GR00T-N1.5 | 20/20 | 18/20 | 19/20 | 12/20 | 18/20 | 15/20 | 17/20 | 13/20 | 15/20 | 16/20 | 10/20 | 14/20 | 12/20 | 85.0 | 73.3 | 60.0 | 76.5 |
| | **DynVLA (Ours)** | **20/20** | **19/20** | **20/20** | **14/20** | **19/20** | **16/20** | **17/20** | **14/20** | **16/20** | **16/20** | **12/20** | **14/20** | **15/20** | **89.3** | **76.7** | **68.3** | **81.5** |
| **Full** | Diffusion Policy | 11/20 | 7/20 | 8/20 | 3/20 | 8/20 | 4/20 | 7/20 | 2/20 | 3/20 | 4/20 | 1/20 | 2/20 | 1/20 | 34.3 | 15.0 | 6.7 | 23.5 |
| | ACT | 15/20 | 10/20 | 12/20 | 5/20 | 11/20 | 8/20 | 8/20 | 4/20 | 6/20 | 7/20 | 2/20 | 5/20 | 3/20 | 49.3 | 28.3 | 16.7 | 36.9 |
| | $\pi_{0.5}$ | 19/20 | 17/20 | 18/20 | 11/20 | 18/20 | 14/20 | 15/20 | 10/20 | 13/20 | 16/20 | 10/20 | 13/20 | 12/20 | 80.0 | 65.0 | 58.3 | 71.5 |
| | GR00T-N1 | 18/20 | 16/20 | 18/20 | 11/20 | 16/20 | 13/20 | 15/20 | 11/20 | 11/20 | 15/20 | 9/20 | 13/20 | 12/20 | 76.4 | 61.7 | 56.7 | 68.5 |
| | GR00T-N1.5 | 20/20 | 18/20 | 19/20 | 13/20 | 18/20 | 15/20 | 17/20 | 13/20 | 15/20 | 17/20 | 10/20 | 14/20 | 12/20 | 85.7 | 75.0 | 60.0 | 77.3 |
| | **DynVLA (Ours)** | **20/20** | **19/20** | **20/20** | **15/20** | **19/20** | **16/20** | **17/20** | **15/20** | **16/20** | **16/20** | **12/20** | **15/20** | **15/20** | **90.0** | **78.3** | **70.0** | **82.7** |

We present the expanded version of the data scaling analysis. The following table details the success counts ($S/20$) for each of the thirteen tasks across the Medium, Large, and Full data scales. This comprehensive breakdown confirms that DynVLA maintains its performance lead in average success rates and category-level results, while achieving strong performance on most individual task logics, particularly in scenarios where imitation-only baselines struggle with error accumulation.

The scaling results indicate that while increasing demonstration data improves the performance of all models, DynVLA shows its largest margin over the strongest baseline in the Small regime, and the gap stabilizes as the data scale increases. This is attributed to the Dynamics Bank's ability to provide structural anchors that prevent the policy from overfitting to noise in the initial few-shot demonstrations. As the scale reaches Full (90+ demonstrations), DynVLA still outperforms

GR00T-N1.5 by 5.4 percentage points in overall average success rate, while consistently achieving higher precision in tasks such as T4 (Orientation) and T6 (Assembly), where execution consistency is critical for success.

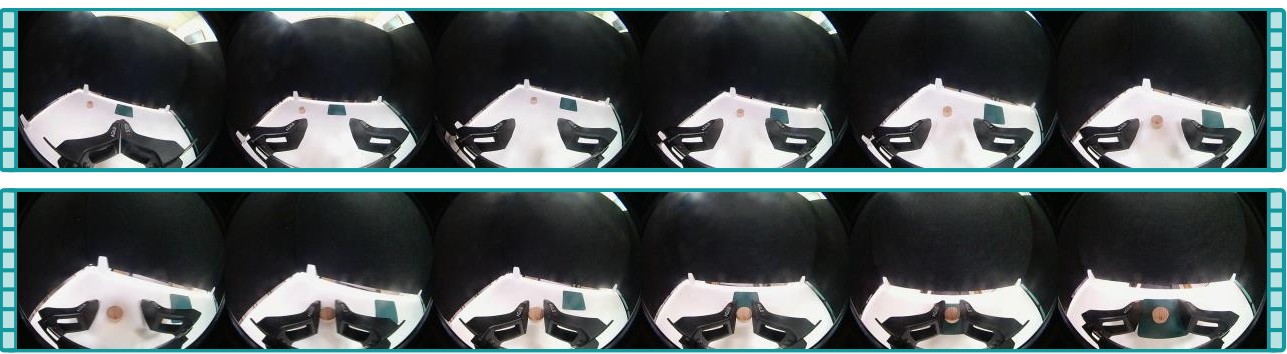

*Figure 7.* (T1) Pick and Place (Fisheye-RGB).

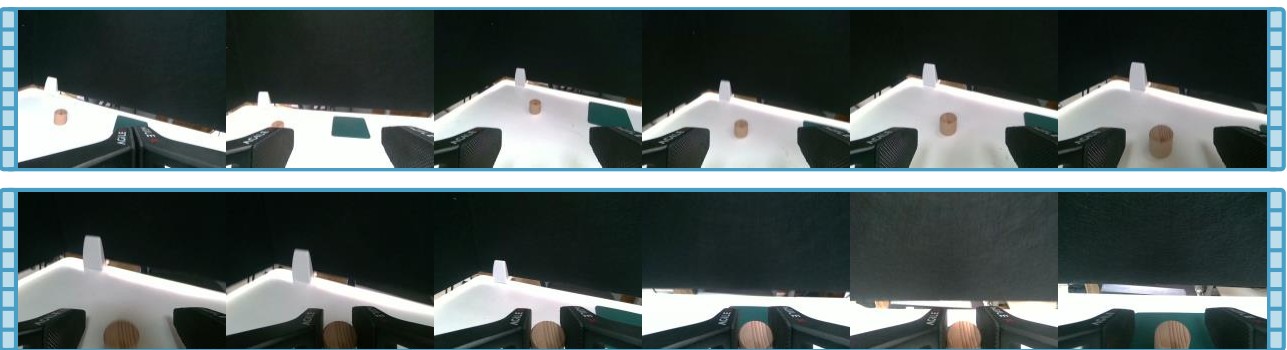

*Figure 8.* (T1) Pick and Place (RealSense-RGB).

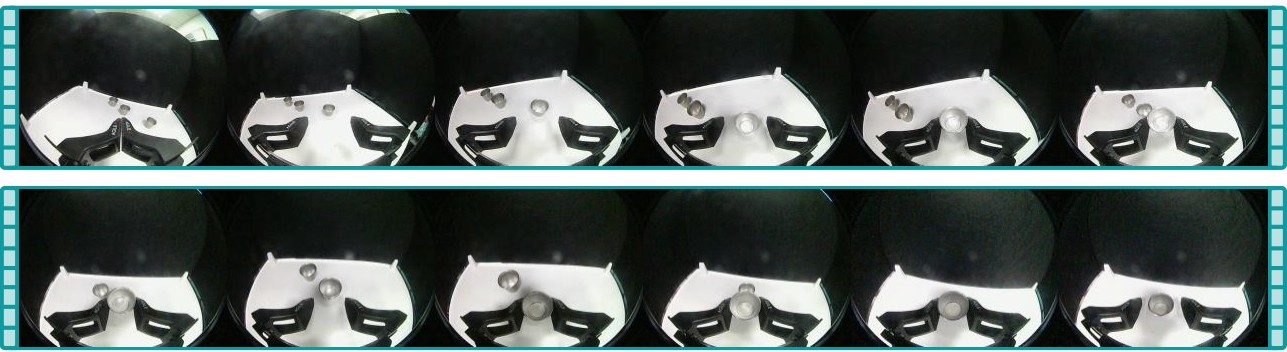

*Figure 9.* (T2) Stacking (Fisheye-RGB).

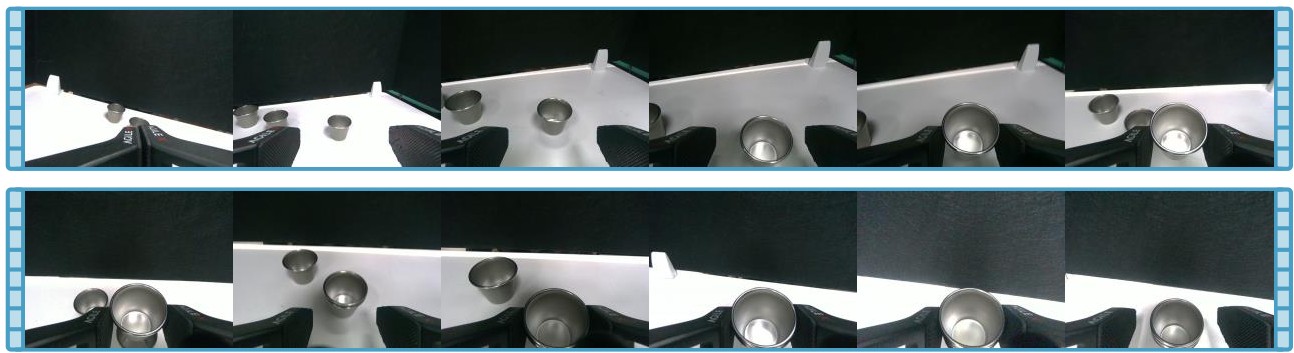

*Figure 10.* (T2) Stacking (RealSense-RGB).

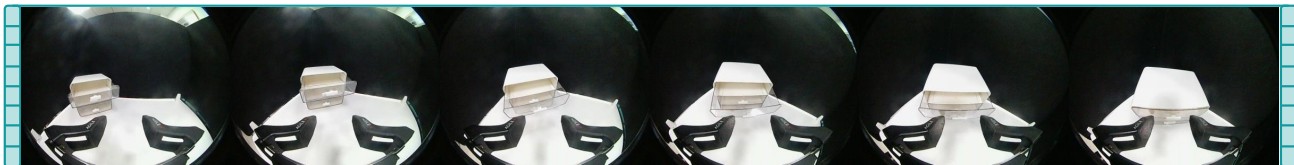

*Figure 11.* (T3) Sliding (Fisheye-RGB).

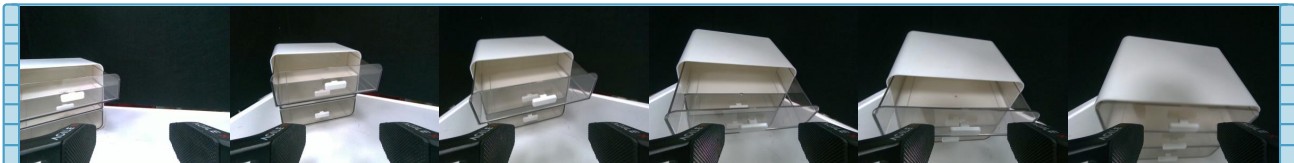

*Figure 12.* (T3) Sliding (RealSense-RGB).

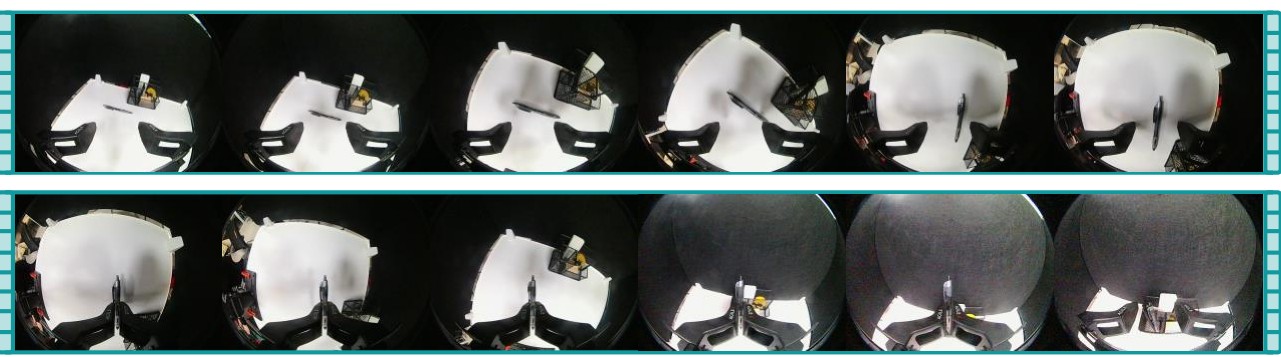

*Figure 13.* (T4) Orientation (Fisheye-RGB).

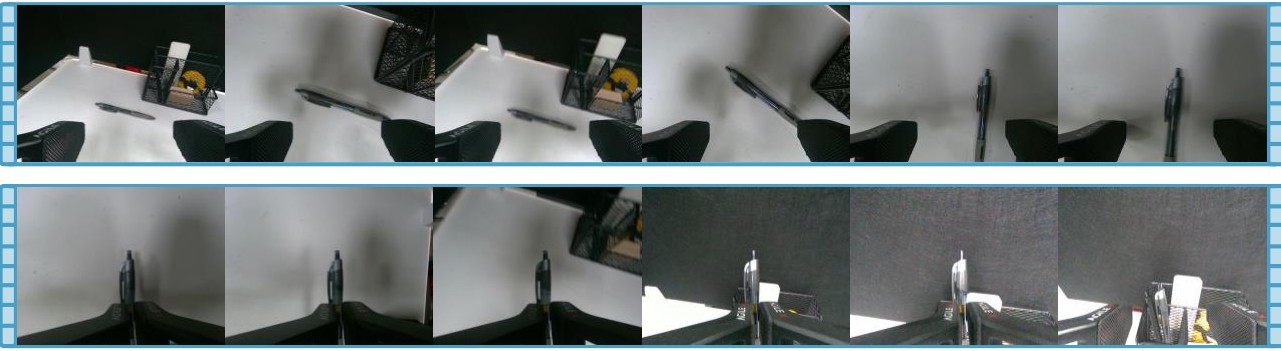

*Figure 14.* (T4) Orientation (RealSense-RGB).

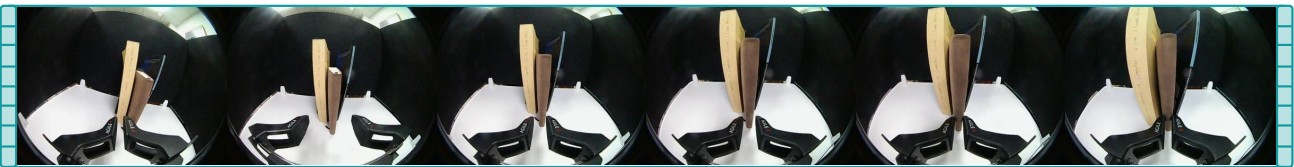

*Figure 15.* (T5) Insertion (Fisheye-RGB).

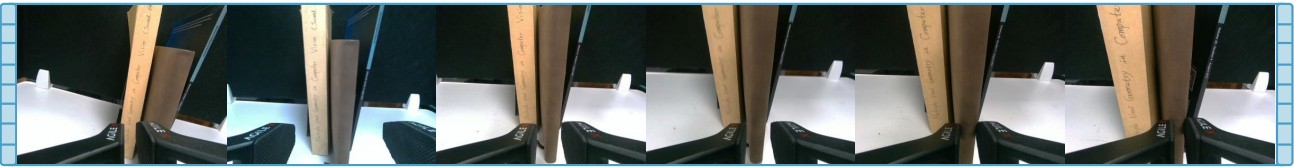

*Figure 16.* (T5) Insertion (RealSense-RGB).

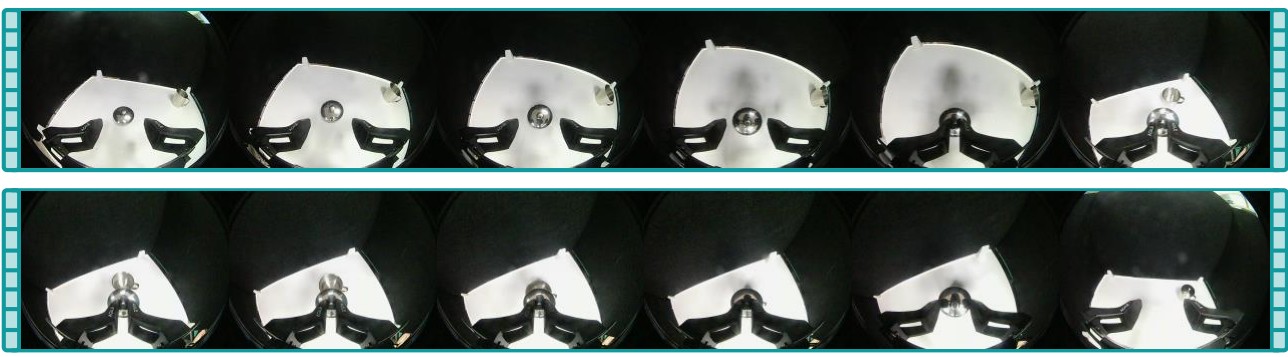

*Figure 17.* (T6) Assembly (Fisheye-RGB).

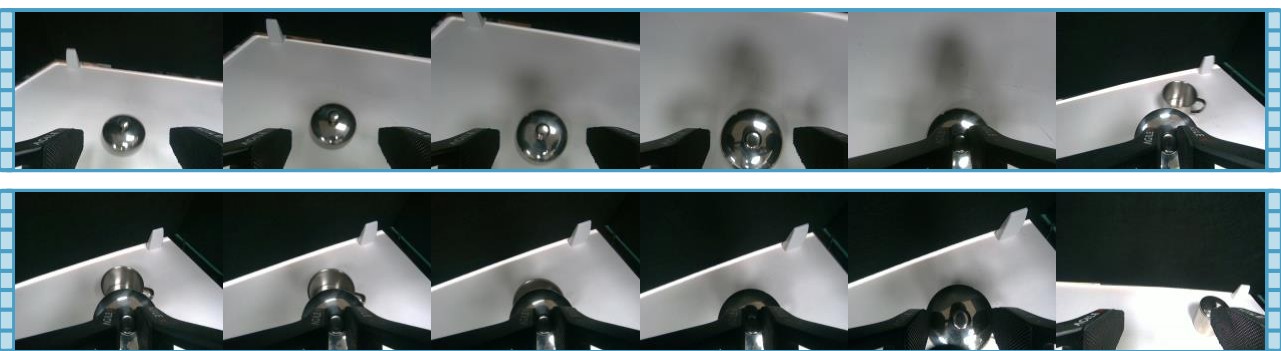

*Figure 18.* (T6) Assembly (RealSense-RGB).

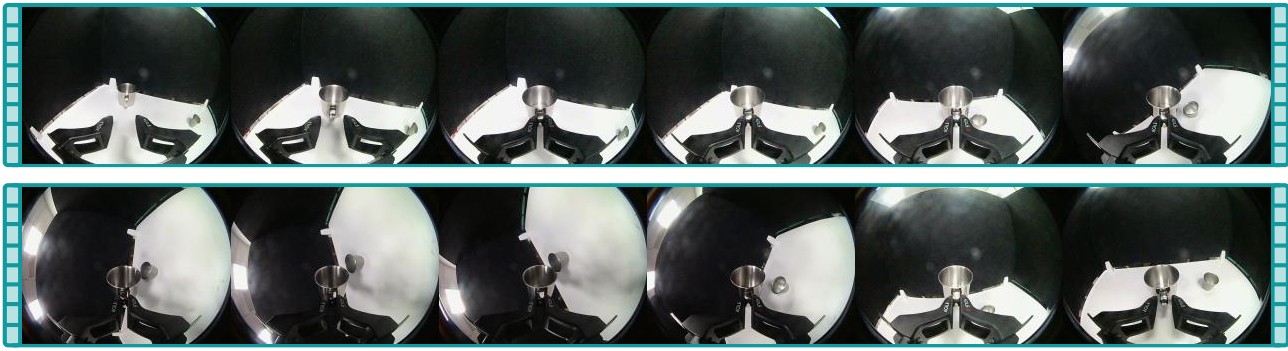

*Figure 19.* (T7) Pouring (Fisheye-RGB).

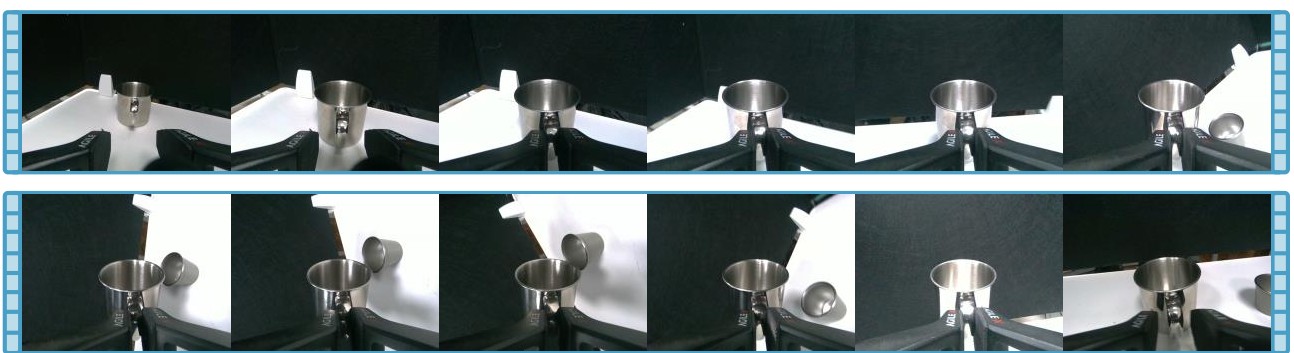

*Figure 20.* (T7) Pouring (RealSense-RGB).

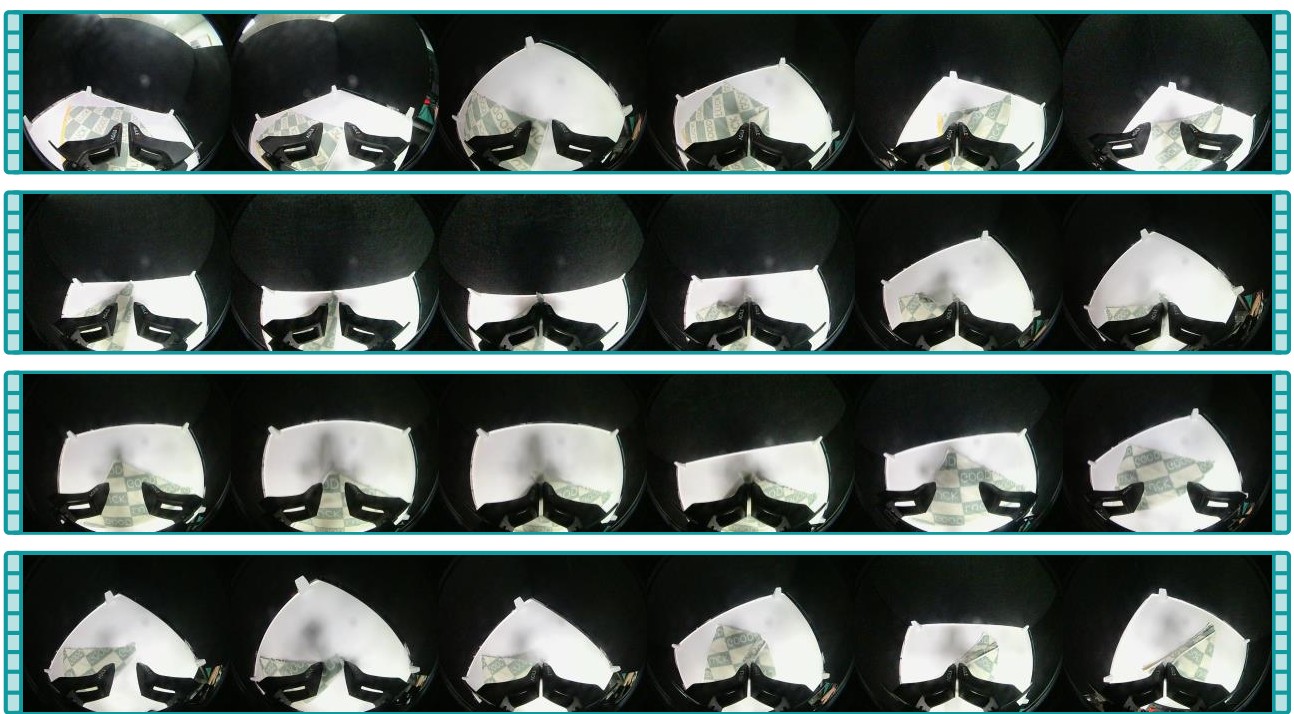

*Figure 21.* (T8) Deformable Manipulation (Fisheye-RGB).

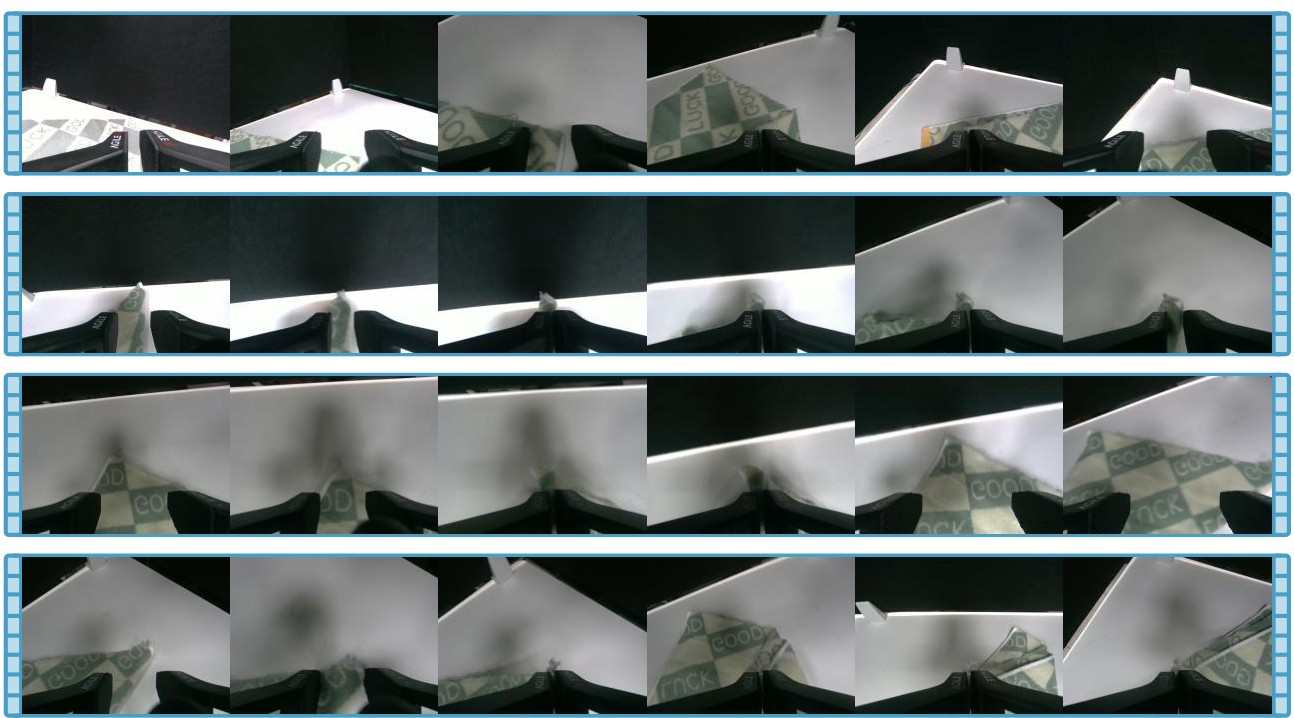

*Figure 22.* (T8) Deformable Manipulation (RealSense-RGB).

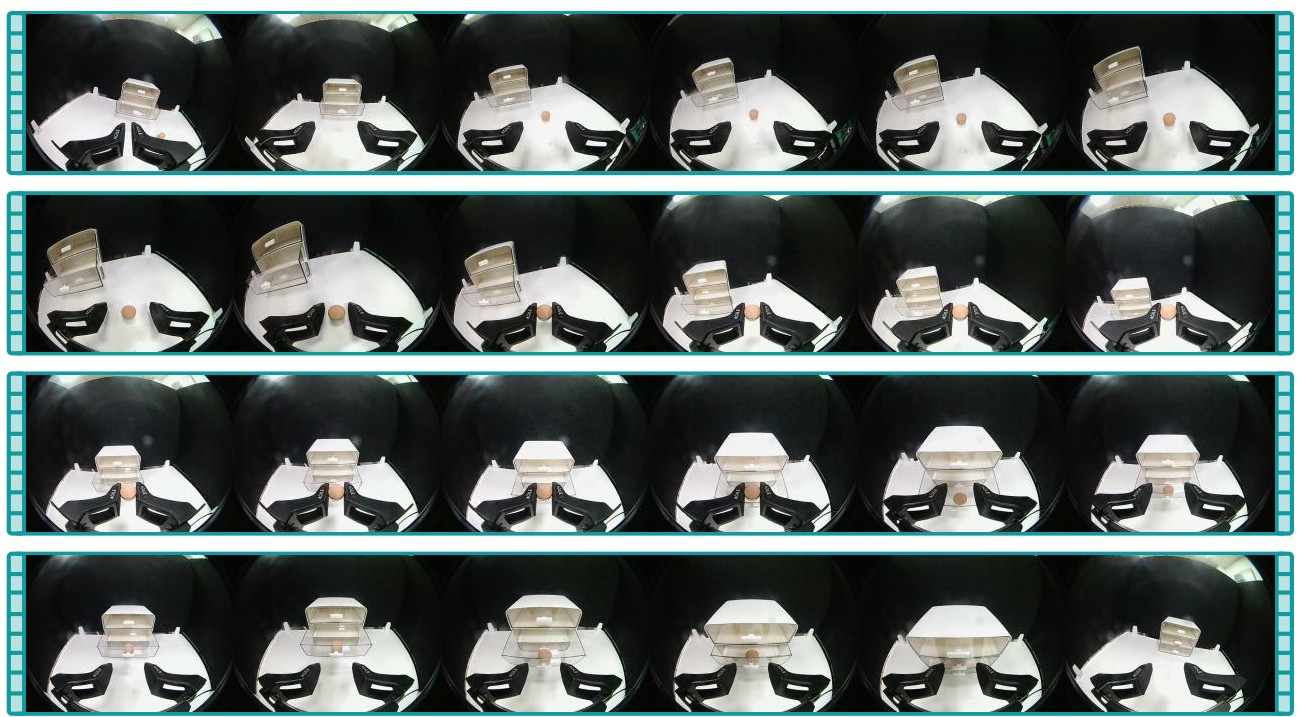

*Figure 23.* (T9) Composite Task (Fisheye-RGB).

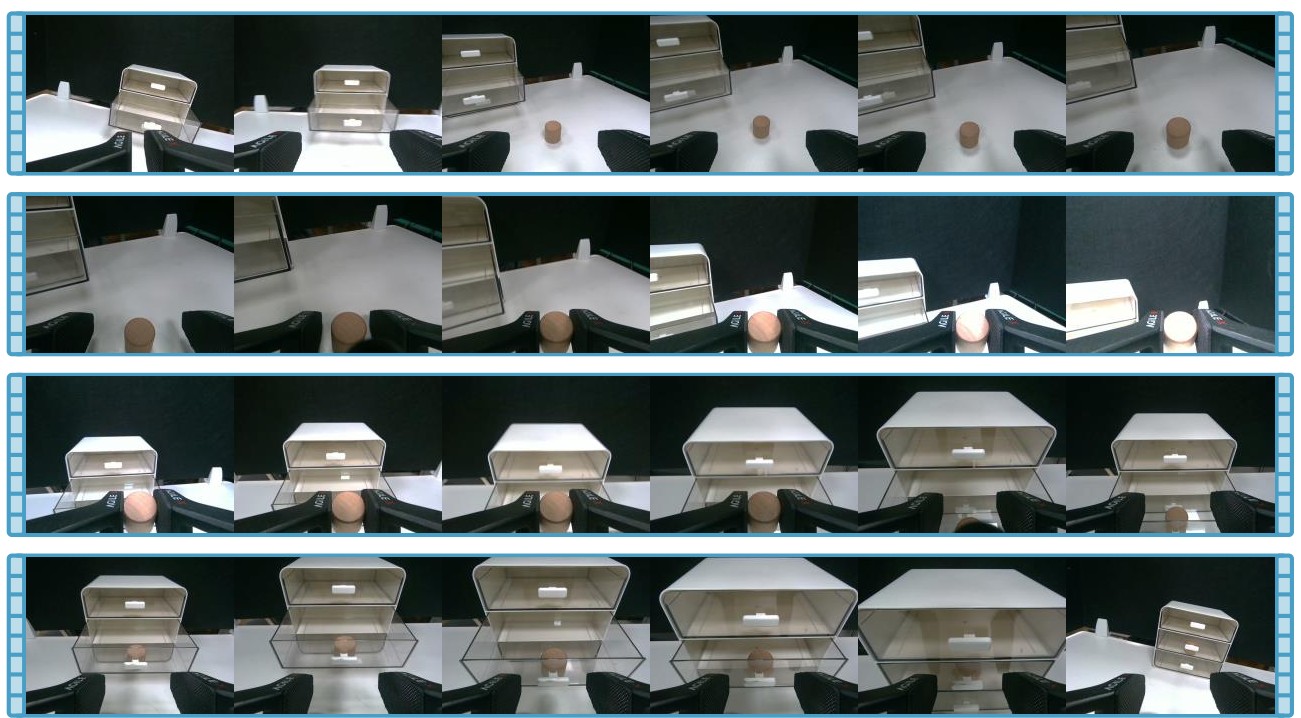

*Figure 24.* (T9) Composite Task (RealSense-RGB).

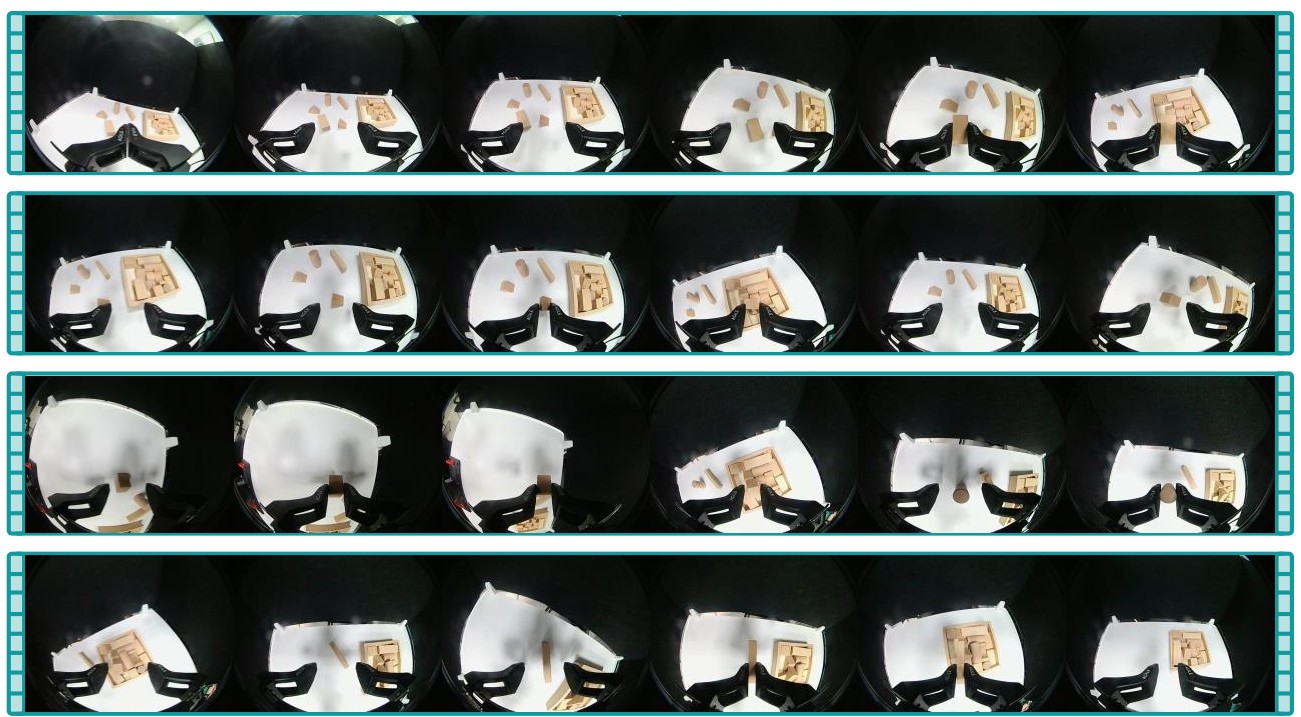

*Figure 25.* (T10) Multi-Object Collection (Fisheye-RGB).

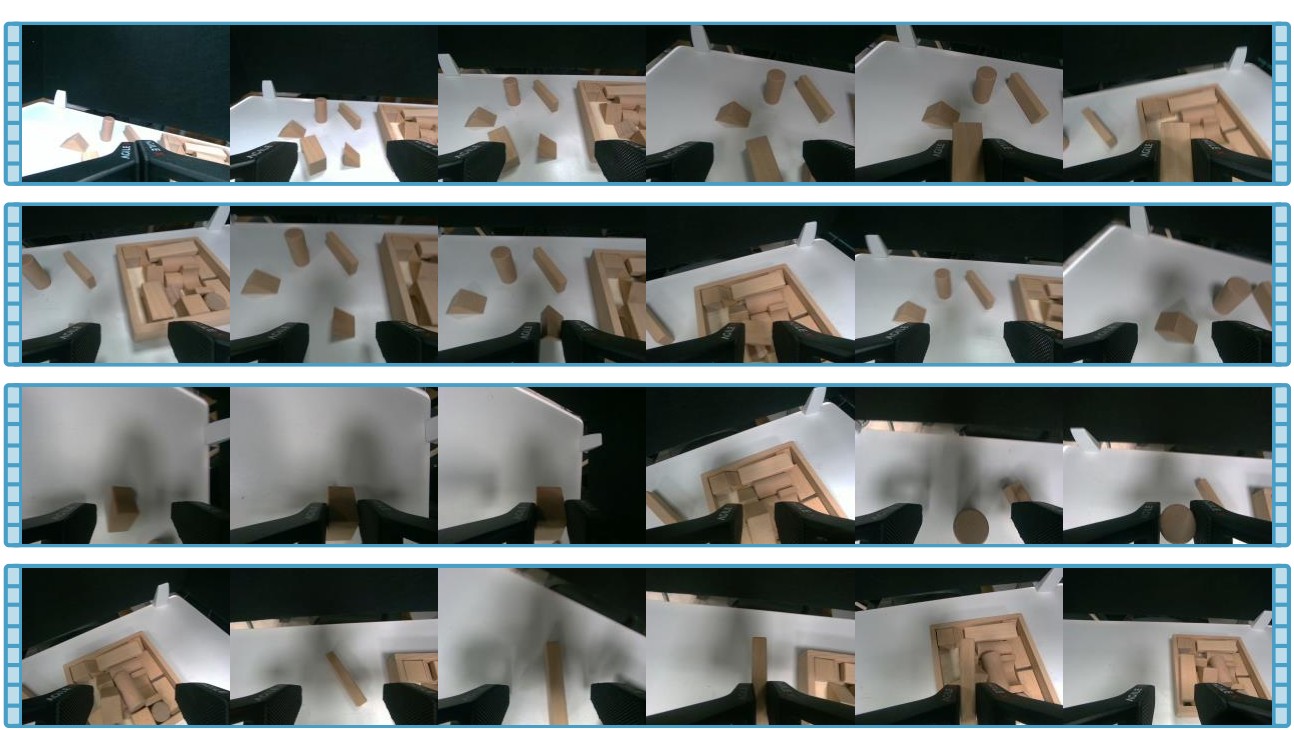

*Figure 26.* (T10) Multi-Object Collection (RealSense-RGB).

