# OpenReview forum: "Motion Dynamics Learning for Few-Shot Embodied Adaptation"
_ICML.cc/2026/Conference — ICML 2026 regular_

### Official Review · Reviewer_928W · 2026-02-15

**Soundness:** 3
**Presentation:** 3
**Significance:** 3
**Originality:** 2
**Overall Recommendation:** 4
**Confidence:** 4

**Summary:**

This paper proposes DynVLA, a dynamics-constrained vision-language-action framework for few-shot embodied adaptation. Instead of relying on frame-wise action imitation, the method reformulates adaptation as trajectory-level motion dynamics modeling. It introduces a Motion Dynamics Mechanism (MDM) that extracts latent physical regimes via flow-matching inversion and temporal contrastive grounding. A Dynamics-Constrained Modeling (DCM) module then projects these representations onto a pretrained dynamics bank to regularize action generation. Evaluated on 13 real-world robotic tasks under short-horizon, long-horizon, and zero-shot settings, DynVLA significantly outperforms state-of-the-art baselines in low-data regimes, demonstrating improved stability and generalization in physical execution.

**Compliance With Llm Reviewing Policy:**

Affirmed.

**Final Justification:**

My issue has been resolved, and I will raise my score to support the acceptance of this paper

**Key Questions For Authors:**

Main Question (Novelty): The authors must clearly articulate how their method differs in design and formulation from the three aforementioned works. Furthermore, they should compare their approach's performance against these prior methods under the same experimental setting. If direct comparison is infeasible due to incompatible setups, a comparison based on real-world success rates on a physical robot would be acceptable. Addressing this question is critical for any potential score improvement.

Minor Issues:

The Dynamics Bank appears to function as a learnable codebook. Is it essentially a form of latent vector quantization (VQ), or equivalent to some form of discrete conditioning?

What is the additional optimization overhead introduced by flow inversion? How long does few-shot adaptation take in practice?

**Limitations:**

The paper does not discuss its limitations. The authors should include a limitations section addressing, for example, failure cases on complex tasks, constraints in real-world robot deployment, or other scenarios where the method underperforms.

**Strengths And Weaknesses:**

Soundness: The paper demonstrates solid methodological soundness by employing a standard Vision-Language-Action (VLA) framework and providing comparisons with established state-of-the-art VLA approaches.

Presentation: The manuscript is well-written, with no noticeable issues in language or structure.

Significance: The work addresses an important challenge in robotic manipulation: teleoperation data collection is costly and labor-intensive. The authors propose a few-shot learning approach to alleviate the data scarcity problem in VLA model training, which holds practical value.

Originality: However, the proposed few-shot paradigm is not entirely novel within the domain of dexterous manipulation. Prior works [1][2][3] have already introduced zero-shot, few-shot, or video-based imitation learning methods that require little to no task-specific data collection while achieving high success rates in real-world manipulation tasks:

[1] ReKep: Spatio-Temporal Reasoning of Relational Keypoint Constraints for Robotic Manipulation. CoRL-LEAP 2024

[2] VITRA: Scalable Vision-Language-Action Model Pretraining for Robotic Manipulation with Real-Life Human Activity Videos. ICRA 2026

[3] UniHM: Unified Dexterous Hand Manipulation with Vision Language Model. ICLR 2026

Overall, I still find the few-shot approach interesting and believe it has potential. If the authors can better highlight the novelty of their method and adequately address the concerns raised below, I would be willing to raise my rating.

---

> ### Author Rebuttal · Authors · 2026-03-30
>
> Thank you for your constructive comments on our work.
>
> 1. (W1&Q1) **Novelty Clarification.**
>
>    Thank you for the comment and for pointing out these relevant works. We will add this discussion into final version.
>
>    We agree that zero-/few-shot manipulation has been explored. In our paper, few-shot adaptation is the target scenario, while our **novelty** lies in how it is formulated and achieved.
>
>    **ReKep (explicit geometric modeling)** relies on explicit relational keypoint constraints and geometric pose solving.
>
>    **VITRA (focuing on scaling law)** addresses data scarcity through large-scale pretraining on human manipulation videos.
>
>    **UniHM (unifying encoding )** introduces VQ-VAE to unify heterogeneous hand kinematics into a discrete action space.
>
>    In contrast, **DynVLA** introduce $r_d$, a **latent representation of dynamics transition**, to **condition action generation** in VLA. This enables adaptation by **learning and reusing transferable dynamics priors**, rather than relying on explicit geometric constraint solving (ReKep), large-scale human-video pretraining (VITRA), or discrete action-space unification (UniHM). **Therefore, our method differs fundamentally in design and in where adaptation is introduced in the pipeline.**
>
>    TABLE 1. Foundational Comparison.
>
>    |Feature|ReKep|VITRA|UniHM|DynVLA|
>    |-|-|-|-|-|
>    |Modeling|Explicit coordinate alignment|Diffusion|Discrete token decode|Flow matching|
>    |Camera view|Global, static|Global, static|Global, static|End-effector, dynamic|
>    |Calibration|required|required|required|not required|
>    |Architecture|Hierarchy|End-to-end|End-to-end|End-to-end|
>
> 2. (W1&Q1) **Emperical Results.**
>
>    We also agree that empirical comparison is essential. We evaluate all methods on 13 real-world tasks and report real-world success rates as a unified metric. As shown in Tables 2 and 3, our method consistently outperforms prior approaches across tasks and platforms, supporting the effectiveness of dynamics-conditioned few-shot adaptation. We also observe that ReKep is prone to keypoint drift, which may cause unsafe end-effector behavior during deployment. The results highlight DynVLA's adaptation capabilities in real-world scenes.
>
>    TABLE 2. Performance Comparison in Single-Arm robot. (Dynamic camera view)
>
>    |Model|Short (/140)|Long (/60)|Zero (/60)|Avg. (%)|
>    |-|-|-|-|-|
>    |ReKep|52|7|17|29.2|
>    |VITRA|38|3|12|20.4|
>    |UniHM|81|13|22|44.6|
>    |DynVLA|114|41|35|73.1|
>
>    TABLE 3. Performance Comparison in Dual-Arm robot. (Static camera view)
>
>    |Model|Short (/140)|Long (/60)|Zero (/60)|Avg. (%)|
>    |-|-|-|-|-|
>    |ReKep|48|18|14|30.8|
>    |VITRA|42|7|11|23.1|
>    |UniHM|98|30|23|58.1|
>    |DynVLA|110|39|34|70.4|
>
>    The Dual-Arm experiment setup is provided in the **Anonymous URL** (in our response to Q2). We will further clarify these differences and comparisons in the final version.
>
> 3. (Q2) **Dynamics Bank Explanation.**
>
>    Following the clarification of $r_d$ (W1&Q1), the bank prototype is **a collection of discrete $r_d$** from diverse pretraining data, providing a comprehensive representation library that conditions DiT to facilitate the reuse of physical priors for new embodiments and tasks. This is further supported by t-sne visualization results.
>
>    **Anonymous URL**: https://drive.google.com/file/d/1XkSIcIBwYM8x7f4GClf0kMVqUpSK_ppG/view?usp=sharing.
>
>    The visualization shows that the learned embeddings are structured by transition evolution rather than purely by task labels: latent states follow continuous paths, and samples with similar transition behaviors remain close even across different tasks. This pattern is **consistent** with the interpretation that $r_d$ captures shared latent physical mechanisms underlying execution dynamics, rather than acting only as a task-specific or generic latent code.
>
> 4. (Q3) **Optimization Overhead and Adaptation Time.**
>
>    Thank you for raising this practical question. The flow inversion process costs a **one-time** running (\~14h). For any subsequent adaptation, the samples (10\~20 shots each task) collection takes \~16h, and the fine-tune costs \~6h in 8 A100 (80G). We provide the latency analysis on different hardwares in Tab 4 and 5 below.
>
>    TABLE 4. Desktop-Grade NVIDIA RTX 4090 (24G).
>
>    |Method|Avg Latency (ms)|Throughput (samples/s)|
>    |-|-|-|
>    |GR00T-N1|61.90|16.16|
>    |ReKep|1267.81|0.79|
>    |VITRA|217.69|4.59|
>    |UniHM|167.05|5.99|
>    |DynVLA|72.54|13.79|
>
>    TABLE 5. Edge Device NVIDIA Jetson Orin (16G).
>
>    |Method|Avg Latency (ms)|Throughput (samples/s)|
>    |-|-|-|
>    |GR00T-N1|283.23|3.53|
>    |ReKep|2281.46|0.44|
>    |VITRA|576.37|1.73|
>    |UniHM|443.21|2.26|
>    |DynVLA|318.52|3.14|
>
> 5. (L1) **Limitation.**
>
>    Thank you for raising this point. In our experiment, we observe a higher manipulation failure rates at the edges of the reachable workspace. We will include a discussion about this observation in the limitation section.

---

> > ### Author Rebuttal · Reviewer_928W · 2026-04-01
> >
> > Thank you for the author's response. I believe the author will discuss the differences between the three articles and supplement with comparative experiments in the final version. Due to these changes, my issue has been resolved, and I will raise my score to support the acceptance of this paper

---

> > > ### Author Response · Authors · 2026-04-01
> > >
> > > Thank you for the valuable feedback and for recognizing our responses. We will include all supplemented experimental results and discussion in the final version.
> > >
> > > We also sincerely appreciate your recognition of our model design novelty and significance.

---

### Official Review · Reviewer_KbX4 · 2026-03-13

**Soundness:** 3
**Presentation:** 3
**Significance:** 3
**Originality:** 3
**Overall Recommendation:** 4
**Confidence:** 1

**Summary:**

This paper proposes DynVLA, a few-shot adaptation framework for vision-language-action models that shifts the objective from frame-level action imitation to trajectory-level motion dynamics modeling. The method has two main components: Motion Dynamics Mechanism (MDM), which uses flow-matching inversion and temporal contrastive grounding to infer a latent dynamics representation, and Dynamics-Constrained Modeling (DCM), which projects that representation onto a pretrained dynamics bank and uses the retrieved prototypes to constrain action generation.

**Compliance With Llm Reviewing Policy:**

Affirmed.

**Final Justification:**

I keep my score and support the acceptance of this paper. The rebuttal fully addressed my concerns, and I really appreciate the authors' detailed responses.

**Key Questions For Authors:**

Listed in the weaknesses.

**Limitations:**

yes.

**Strengths And Weaknesses:**

Strengths:
-  The shift from standard behavioral cloning to flow-matching inversion for trajectory-level dynamics modeling is an interesting and well-motivated approach for VLAs.
-  The proposed method demonstrates strong empirical results: 19% improvement on success rate across 13 diverse real-world tasks.
- Well-written theroetical proof is proposed for the method.

Weaknesses:
- Is the inversion in Eq. (4) well-posed? Could multiple distinct $r_d$ values produce similarly low inversion loss for the same trajectory? If so, how does the method avoid ambiguity or instability in the learned dynamics representation? Could author please clarify?

---

> ### Author Rebuttal · Authors · 2026-03-30
>
> Thank you for the valuable feedback on our work.
>
> 1. (W1&Q1) **Explanation of Well-Posed Optimization.**
>
>    We deeply appreciate the reviewer for raising this critical question. Our design of **Gounding via Contrastive Learning** (Section 4.1.2) is exactly motivated to address the issue of well-posed optimization in FM inversion (Equation 4). Without this design, the inversion is ill-posed because it is a randomized divergent optimization allowing multiple distinct $r_d$ to map to the same trajectory.  **Our contrastive grounding** resolves this by explicitly regularizing the latent space, pulling representations of dynamically similar trajectories together and pushing dissimilar ones apart. This drives the optimization to **converge toward a unique $r_d$** that strictly aligns with the underlying physical dynamics.
>
>    To demonstrate how our method effectively resolves this ambiguity and ensures a well-posed representation, we conduct comprehensive experiments including **qualitative visualization** (E1), **quantitative ablation** (E2) and **cross-embodiment consistency** (E3).
>
> 2. (E1) **Qualitative Visualization.**
>
>    We additionally provide a t-SNE visualization of the learned $r_d$ across diverse manipulation tasks in  **anonymous URL**: https://drive.google.com/file/d/1XkSIcIBwYM8x7f4GClf0kMVqUpSK_ppG/view?usp=sharing
>
>    The visualization shows that the learned embeddings are structured by transition evolution rather than purely by task labels: latent states follow continuous paths, and samples with similar transition behaviors remain close even across different tasks.  Thus there is consistency across similar trajectories. This pattern is consistent with the interpretation that $r_d$ captures shared latent physical mechanisms underlying execution dynamics, rather than acting only as a task-specific or generic latent code.
>
> 3. (E2) **Quantitative Ablation.**
>
>    The ablation study conducted across the 13 tasks (detailed in main paper) demonstrates that Grounding via Contrastive Learning effectively resolves the ambiguity or instability in the learned dynamics representation.
>
>    TABLE 1. Ablation  Study of Contrastive Learning.
>    |Config|Short (%)|Long (%)|Zero (%)| Avg. (%)|
>    |-|-|-|-|-|
>    |w/o contrastive learning|61.4|41.7|38.3|51.5|
>    |DynVLA|81.4|68.3|58.3|73.1|
>
> 4. (E3) **Cross-Embodiment Consistency.**
>
>    To verify the consistency of  $r_d$ for similar trajectories across different embodiments,  we transfer the pretrained model setup to a new robot (shown in the same anonymous URL). Specifically, we collect 10-20 demonstrations for each task (except zero-shot tasks) in entirely new scenes and compare the few-shot performance against baseline models. The results detailed in Table 1 confirm that **$r_d$ keeps consistent for  similar trajectories even across different embodiments.**
>
>    TABLE 2. Performance Comparison. (7 short-horizon tasks, 3 long-horizon tasks, 3 zero-shot tasks, 20 tests for a task)
>
>    |Model|Short (/140)|Long (/60)|Zero (/60)|Avg. (%)|
>    |-|-|-|-|-|
>    |Diffusion Policy|25|5|0|11.5|
>    |ACT|38|10|2|19.2|
>    |$\pi_{0.5}$|78|21|23|46.9|
>    |GR00T-N1|70|19|20|41.9|
>    |DynVLA|110|39|34|70.4|
>
> To make this design rationale clearer to the readers, we will add it to the final version to explicitly discuss the issue and highlight how our contrastive learning module effectively resolves it.

---

> > ### Author Rebuttal · Reviewer_KbX4 · 2026-04-03
> >
> > Thanks for the detailed responses. My concerns have been fully addressed, and I will keep my score to support the acceptance.

---

> > > ### Author Response · Authors · 2026-04-03
> > >
> > > Thank you for the valuable feedback and for recognizing our responses. We will include all supplemented experimental results and discussion in the final version.
> > >
> > > We also sincerely appreciate your recognition of our model design novelty and significance.

---

### Official Review · Reviewer_U1qD · 2026-03-14

**Soundness:** 2
**Presentation:** 3
**Significance:** 3
**Originality:** 2
**Overall Recommendation:** 4
**Confidence:** 4

**Summary:**

This paper proposes DynVLA, a few-shot adaptation framework for Vision-Language-Action (VLA) models. It adapts pretrained VLA models via trajectory-level motion dynamics modeling, which is more efficient and transferable than direct action imitation. The system consists of two components: MDM, which extracts a latent dynamics representation from few-shot demonstrations via flow-matching inversion, and DCM, which grounds online action generation in a pretrained prototype bank via soft-attention retrieval. Experiments on 13 self-collected     real-world manipulation tasks show consistent gains over several strong baselines under few-shot settings.

**Compliance With Llm Reviewing Policy:**

Affirmed.

**Key Questions For Authors:**

1. Can you clarify the concept of "physical regimes" or "physical modes"?
2. Does the bank prototypes represents anything clear in the real world?
3. How is the MDM inversion different from the parameter identification in the classic control theory? A direct comparison of these study and extension of current related work will clarify this point.
4. What is the wall clock time and inference time overhead of this system as a whole?

**Limitations:**

This paper does not directly include a limitation section. The limitation of this work lies in the ambiguity of the concepts at the heart of the paper. The user should clarify the concept of physical regimes and what the method truly learns and adapts to.

**Strengths And Weaknesses:**

### Strengths:
1. The problem of transferring physical modes from the training domain to the testing domain is a valuable and plausible problem.
2. The authors have designed the MDM and DCM for learning and transferring such knowledge into another domain.
3. The real-world experiment is sufficient.


### Weaknesses:
1. The core concept of this paper, "physical regimes", "latent physical regimes," or "physical modes," is not defined in the paper.  Physical modes, such as velocity, speed, friction, and other dynamics, can be characterized through system identification and control.  However, no separate study, case study, or visualization has been conducted to clarify the exact meaning of these concepts in the paper.
2. The experiments are all conducted on the dataset the paper proposes, there are no experiments on other open-source benchmarks.

---

> ### Author Rebuttal · Authors · 2026-03-30
>
> Thank you for the constructive feedback on our work.
>
> 1. (W1&Q1) **Explanation of Physical Regimes.**
>
>    We agree that our previous description should be made more precise. The concept of "physical regimes" or "physical modes" denotes **the underlying collection of physical properties that govern how a trajectory envolves.** We will **refine this phrasing** in the final manuscript.
>
>    We do not claim that $r_d$ is directly equivalent to any explicit physical quantity, but that it provides a latent parameterization of observable physical dynamics. Specifically, we model $r_d$ as a latent representation of dynamics transition. $r_d$ abstracts physically meaningful factors underlying execution dynamics in a high-dimensional latent space. This allows the model to capture nonlinear and coupled physical variations that are difficult to express with hand-crafted variables, while remaining grounded in observable motion behaviors such as smoothness, speed profile, and feasibility. It is in line with the latent representation theory such as JEPA [1, 2]. Thus, our latent representation $r_d$ in VLA systems offers a highly practical and scalable solution for deployment.
>
>    [1] Dawid, Anna, and Yann LeCun. "Introduction to latent variable energy-based models: a path toward autonomous machine intelligence." Journal of Statistical Mechanics: Theory and Experiment (2024).
>
>    [2] Assran, Mido, et al. "V-JEPA 2: Self-supervised video models enable understanding, prediction and planning." arXiv:2506.09985 (2025).
>
> 2. (Q2) **Explanation of Bank Prototypes.**
>
>    Following the abstraction of physical regimes and clarification of $r_d$, the bank prototype that we construct is **a collection of $r_d$** from diverse pretraining data, providing a comprehensive representation library that facilitates the reuse of physical priors for new embodiments and novel tasks. This is supported by the additional experiments below.
>
> 3. (Q3) **MDM Inversion & Param Identification.**
>
>    Thanks for this point. We will add it to related work of final version.
>
>    |Feature|Classic Parameter Identification [3]|MDM Inversion (Ours)|
>    |-|-|-|
>    |Target|Explicit physical variables (e.g., friction)|Latent dynamics embedding ($r_d$)|
>    |Modeling|Rigid analytical equations|Flow-matching|
>    |Calibration|Tedious|No required|
>    |Scalability|Specific|General|
>    |Time cost|High|Low|
>
>    [3] Ljung, Lennart. "System identification." Signal analysis and prediction (1998).
>
> 4. (W1) **Visualization.**
>
>    We additionally provide a t-SNE visualization of the learned $r_d$ across diverse manipulation tasks in  **anonymous URL**: https://drive.google.com/file/d/1XkSIcIBwYM8x7f4GClf0kMVqUpSK_ppG/view?usp=sharing
>
>    The visualization shows that the learned embeddings are structured by transition evolution rather than purely by task labels: latent states follow continuous paths, and samples with similar transition behaviors remain close even across different tasks. This pattern is consistent with the interpretation that $r_d$ captures shared latent physical mechanisms underlying execution dynamics, rather than acting only as a task-specific or generic latent code.
>
> 5. (W1) **Cross-Embodiment Transferability.**
>
>    To verify the cross-embodiment generalization, we transfer the pretrained model to a dual-arm robot (the URL above), and compare the few-shot (10~20) performance. The results detailed in Table 1 confirm that $r_d$ **captures underlying physical regimes**  across embodiments.
>
>    TABLE 1. Performance Comparison. (7 short-horizon, 3 long-horizon, 3 zero-shot, 20 tests each task)
>
>    |Model|Short (/140)|Long (/60)|Zero (/60)|Avg. (%)|
>    |-|-|-|-|-|
>    |Diffusion Policy|25|5|0|11.5|
>    |ACT|38|10|2|19.2|
>    |$\pi_{0.5}$|78|21|23|46.9|
>    |GR00T-N1|70|19|20|41.9|
>    |DynVLA|110|39|34|70.4|
>
>    TABLE 2. Ablation Study of Bank Prototypes.
>    |Configuration|Short (/140)|Long (/60)|Zero (/60)|Avg. (%)|
>    |-|-|-|-|-|
>    |w/o Bank Prototypes|92|27|29|56.9|
>    |DynVLA|110|39|34|70.4|
>
> 6. (W2) **Open-Source Benchmark.**
>
>    We conduct experiments at mainstream open-source benchmark LIBERO.
>
>    TABLE 3. Performance Comparison in LIBERO (%).
>
>    |Method|Spatial|Object|Goal|LIBERO-10|Avg.|
>    |-|-|-|-|-|-|
>    |$\pi_{0.5}$|98.8|98.2|98.0|92.4|96.9|
>    |GR00T N1|94.4|97.6|93.0|90.6|93.9|
>    |DynVLA|97.3|99.5|96.7|94.8|97.1|
>
> 7. (Q4) **Time Analysis.**
>
>    We provide the latency analysis on a desktop-grade NVIDIA RTX 4090 GPU (24G) and an NVIDIA Jetson Orin (16G) edge device.
>
>    TABLE 4. Analysis in RTX 4090.
>    |Method|Avg Latency (ms)|Throughput (samples/s)|
>    |-|-|-|
>    |$\pi_{0.5}$|193.93|5.16|
>    |GR00T-N1|61.90|16.16|
>    |DynVLA|72.54|13.79|
>
>    TABLE 5. Analysis in Jetson Orin  ($\pi_{0.5}$ is out of memory).
>    |Method|Avg Latency (ms)|Throughput (samples/s)|
>    |-|-|-|
>    |GR00T-N1|283.23|3.53|
>    |DynVLA|318.52|3.14|
>
> 8. (L1) **Limitations.**
>
>    Thank you for raising this point. We will add the clarification and discussion into the final version.

---

> > ### Author Rebuttal · Reviewer_U1qD · 2026-04-06
> >
> > My concerns have been adequately addressed.

---

> > > ### Author Response · Authors · 2026-04-06
> > >
> > > Thank you for the valuable feedback and for recognizing our responses. We will include all supplemented experimental results and discussion in the final version.
> > >
> > > We also sincerely appreciate your recognition of our model design novelty and significance.

---

### Official Review · Reviewer_zuhm · 2026-03-17

**Soundness:** 2
**Presentation:** 4
**Significance:** 2
**Originality:** 3
**Overall Recommendation:** 4
**Confidence:** 4

**Summary:**

DynVLA tackles few-shot robot learning by shifting from simple action imitation to modeling motion dynamics of trajectories. It introduces (1) MDM, which extracts latent dynamics from demonstrations via flow-matching inversion, and (2) DCM, which constrains action generation using a pretrained dynamics bank of motion patterns. This lets the model reuse and interpolate physical behaviors rather than relearn them from scratch. With only 10–20 demos, DynVLA achieves significantly better performance across 13 real-world tasks.

**Compliance With Llm Reviewing Policy:**

Affirmed.

**Final Justification:**

All my concerns and questions are well addressed in the rebuttal.

**Key Questions For Authors:**

1. Does the learned dynamics representation $r_d$ generalize across different embodiments? The paper frames $r_d$ ​as capturing reusable “motion dynamics” that are transferable across tasks. However, it is unclear whether these dynamics are embodiment-agnostic (e.g., capturing task-level physical structure), or implicitly tied to a specific robot morphology and control space.

2. Can you provide visualizations or controlled studies to show that $r_d$  meaningfully corresponds to interpretable dynamics? Is there consistency across similar trajectories?

**Limitations:**

yes

**Strengths And Weaknesses:**

### Strength

1. The paper is well-structured with a clear pipeline: motivation → MDM → DCM → experiments.

2. Figures (e.g., overview + dynamics bank) effectively illustrate the intuition behind “dynamics grounding”.

3. The idea of reusing motion dynamics instead of re-learning policies is practically meaningful for robotics, where data is expensive.

### Weaknesses

1. The central claim that the latent variable $r_d$ captures interpretable physical dynamics (such as velocity, smoothness, or feasibility) is not rigorously validated. While $r_d$is introduced as a latent conditioning variable inferred via flow-matching inversion (Equation 4), there is no explicit supervision to enforce what it should encode. The interpretation relies on a theoretical decomposition into "task prior + dynamic correction" (Equation 9), which remains heuristic and lacks empirical verification. Moreover, despite claims that $r_d$ captures "latent physical regimes," the paper provides no visualization, disentanglement analysis, or probing experiments to substantiate this. Consequently, it remains unclear whether $r_d$genuinely encodes physical properties or merely functions as a generic latent embedding that improves optimization.

2. The method exhibits significant sensitivity to architectural choices, undermining its robustness. Codebook size has a narrow optimal range (K=1024), with smaller values lacking expressivity and larger values (K=4096) causing overfitting that degrades zero-shot performance (Table 5). Similarly, constraint injection depth requires precise tuning (exactly N=6 out of 12 layers), as deviations of just a few layers in either direction substantially harm performance (Table 6). This fragility suggests DynVLA is not a plug-and-play solution but requires careful, task-specific hyperparameter optimization, raising concerns about its transferability to new settings without extensive re-tuning.

---

> ### Author Rebuttal · Authors · 2026-03-30
>
> Thank you for your constructive comments on our work.
>
> 1. (W1&Q1) **Explanation of $r_d$.**
>
>    Thank you for this important question. We do not claim that $r_d$ is directly equivalent to any explicit physical quantity, but that it provides a latent parameterization of observable physical dynamics. Specifically, we model $r_d$ as a latent representation of dynamics transition. $r_d$ abstracts physically meaningful factors underlying execution dynamics in a high-dimensional latent space. This allows the model to capture nonlinear and coupled physical variations that are difficult to express with hand-crafted variables, while remaining grounded in observable motion behaviors such as smoothness, speed profile, and feasibility. It is in line with the latent representation theory such as JEPA [1, 2]. Thus, our latent representation $r_d$ in VLA systems offers a highly practical and scalable solution for deployment.
>
>    As Equations (4) and  (10) in Section 4.1, $r_d$ is **explicitly supervised** by the **inversion objective**:
>    $$
>    r\_d^\star=\arg\min_{r_d}\mathcal{L}_{\mathrm{fm\text{-}inv}}(r_d) = \arg\min\_{r\_d} \mathbb{E}\_{\tau,\epsilon}
>    \left\| \pi\_\theta ( f\_c, A\_H^\tau, \tau, r\_d ) - (\epsilon - A\_H) \right\|_2^2.
>    $$
>
>    In this supervision, $r_d$ encodes the **dynamic correction** required for transfer, corresponding **Equation (9)** which is  derived in main paper. This theoretical perspective is **further supported by the following experiments**.
>
>    [1] Dawid, Anna, and Yann LeCun. "Introduction to latent variable energy-based models: a path toward autonomous machine intelligence." Journal of Statistical Mechanics: Theory and Experiment (2024).
>
>    [2] Assran, Mido, et al. "V-JEPA 2: Self-supervised video models enable understanding, prediction and planning." arXiv:2506.09985 (2025).
>
> 2. (W1&Q2) **Visualization.**
>
>    We additionally provide a t-SNE visualization of the learned $r_d$ across diverse manipulation tasks in  **anonymous URL**: https://drive.google.com/file/d/1XkSIcIBwYM8x7f4GClf0kMVqUpSK_ppG/view?usp=sharing
>
>    The visualization shows that the learned embeddings are structured by transition evolution rather than purely by task labels: latent states follow continuous paths, and samples with similar transition behaviors remain close even across different tasks.  Thus there is consistency across similar trajectories. This pattern is consistent with the interpretation that $r_d$ captures shared latent physical mechanisms underlying execution dynamics, rather than acting only as a task-specific or generic latent code.
>
> 3. (W1&Q1&Q2) **Cross-Embodiment Transferability.**
>
>    To verify the cross-embodiment generalization of $r_d$, we transfer the pretrained model from a single-arm setup to a dual-arm robot (shown in the same anonymous URL). Specifically, we collect 10-20 demonstrations for each task (except zero-shot tasks) in entirely new scenes and compare the few-shot performance against baseline models. The results detailed in Table 1 confirm that $r_d$ captures underlying physical regimes and **generalizes across different embodiments.**
>
>    TABLE 1. Performance Comparison. (7 short-horizon tasks, 3 long-horizon tasks, 3 zero-shot tasks, 20 tests for a task, detailed in main paper)
>
>    |Model|Short (/140)|Long (/60)|Zero (/60)|Avg. (%)|
>    |-|-|-|-|-|
>    |Diffusion Policy|25|5|0|11.5|
>    |ACT|38|10|2|19.2|
>    |$\pi_{0.5}$|78|21|23|46.9|
>    |GR00T-N1|70|19|20|41.9|
>    |DynVLA|110|39|34|70.4|
>
> 4. (W2) **Sensitivity to Architectural Choices.**
>
>    We thank for raising this important point. To verify the sensitivity to architectural choices, we conduct ablation studies on the dual-arm robot. As shown in Tables 2 and 3, the hyperparameter settings exhibit consistent trends across different embodiments, uniformly achieving optimal performance with a Codebook Size of 1024 and Depth N=6. This demonstrates that **the hyperparameter settings generalize across different embodiments.** Relative to the baselines, the drop is marginal and the method remains objectively competitive. Furthermore, the choice of middle layers is standard in advanced studies [3,4]. Thus, we respectfully suggest that it is a general and robust setting rather than a narrow band.
>
>    TABLE 2. Ablation of Codebook.
>
>    |Codebook Size K|Short (%)|Long (%)|Zero (%)|Avg. (%)|
>    |-|-|-|-|-|
>    |512|75.0|58.3|51.7|65.8|
>    |1024 (Default)|78.6|65.0|56.7|70.4|
>    |2048| 77.1|63.3|55.0|68.8|
>    |4096| 76.4|61.7|55.0|68.1|
>
>    TABLE 3. Ablation of Depth.
>
>    |Depth N|Short (%)|Long (%)|Zero (%)|Avg. (%)|
>    |-|-|-|-|-|
>    |2|52.9|31.7|30.0|42.7|
>    |4|70.0|51.7|45.0|59.2|
>    |6 (Default)|78.6|65.0|56.7|70.4|
>    |8|73.6|58.3|50.0|64.6|
>    |10|51.4|41.7|43.3|47.3|
>
>    [3] Skean, Oscar, et al. "Layer by Layer: Uncovering Hidden Representations in Language Models." ICML (2025).
>
>    [4] Kaduri, Omri, Shai Bagon, and Tali Dekel. "What's in the Image? A Deep-Dive into the Vision of Vision Language Models." CVPR (2025).

---

> > ### Author Rebuttal · Reviewer_zuhm · 2026-04-03
> >
> > Thanks for the detailed rebuttal provided. I will update my score accordingly.

---

> > > ### Author Response · Authors · 2026-04-03
> > >
> > > Thank you for the valuable feedback and for recognizing our responses. We will include all supplemented experimental results and discussion in the final version.
> > >
> > > We also sincerely appreciate your recognition of our model design novelty and significance.

---

### Decision · Program_Chairs · 2026-04-30

**Decision:**

Accept (regular)

**Comment:**

This paper proposes DynVLA, a few-shot adaptation framework for VLA models that conditions action generation on latent dynamics representations extracted via flow-matching inversion and grounded via temporal contrastive learning, with a pretrained Dynamics Bank providing reusable motion priors.

Strengths:

- Strong real-world results: 19% average success rate improvement over baselines (including pi0.5, GR00T-N1, SmolVLA) across 13 tasks with only 10-20 demonstrations
- Cross-embodiment transfer demonstrated in rebuttal (single-arm to dual-arm robot, 70.4% avg)
- Competitive on LIBERO benchmark (97.1%), confirming results are not specific to the custom task suite
- Practical inference latency (72.54ms on RTX 4090, comparable to GR00T-N1)
- Clean ablation showing both MDM (contrastive grounding) and DCM (dynamics bank) contribute substantially
- Thorough rebuttal adding dual-arm experiments, LIBERO results, comparison with ReKep/VITRA/UniHM, latency analysis, and t-SNE visualization

Weaknesses (minor):

- The "physical regimes" / "dynamics modes" terminology overstates what r_d captures. The authors acknowledged it is a learned latent representation, not interpretable physical quantities. The theoretical justification (Eq. 7-9 Bayes decomposition) is heuristic.
- Individual components are standard ML building blocks (FM inversion, InfoNCE, soft-attention codebook retrieval). The contribution is the combination for the VLA few-shot adaptation.
- Generalization is validated within tabletop manipulation only. Cross-environment or cross-object-category transfer is not tested.

I recommend accepting the paper, given unanimous support.

Revision requests for camera-ready:

- Refine the "physical regimes" / "latent physical regimes" terminology to more accurately describe what r_d represents (a trajectory-conditioned latent code, not explicit physics)
- Include the cross-embodiment experiments (dual-arm), LIBERO results, and ReKep/VITRA/UniHM comparisons from the rebuttal
- Add a limitations section discussing the scope of generalization (tabletop manipulation, similar object categories) and the workspace-edge failure mode
- Include the t-SNE visualization and latency analysis